# Retrieval of snow freeboard of Antarctic sea ice using waveform fitting of CryoSat-2 returns

Steven W. Fons[1,2,3] and Nathan T. Kurtz[3]

[1]Department of Atmospheric and Oceanic Sciences, University of Maryland, College Park, Maryland, USA
[2]Earth System Science Interdisciplinary Center (ESSIC), University of Maryland, College Park, Maryland, USA
[3]Cryospheric Sciences Laboratory, NASA Goddard Space Flight Center, Greenbelt, Maryland, USA

*Correspondence to:* Steven W. Fons (steven.w.fons@nasa.gov)

**Abstract** In this paper we develop a CryoSat-2 algorithm to retrieve the surface elevation of the air-snow interface over Antarctic sea ice. This algorithm utilizes a two-layer physical model that accounts for scattering from a snow layer atop sea ice as well as scattering from below the snow surface. The model produces waveforms that are fit to CryoSat-2 level 1B data through a bounded trust region least squares fitting process. These fit waveforms are then used to track the air-snow interface and retrieve the surface elevation at each point along the CryoSat-2 ground track, from which the snow freeboard is computed. To validate this algorithm, we compare retrieved surface elevation measurements and snow surface radar return power levels with those from Operation IceBridge, which flew along a contemporaneous CryoSat-2 orbit in October 2011 and November 2012. Average elevation differences (standard deviations) along the flight lines (IceBridge Airborne Topographic Mapper (ATM) – CryoSat-2) are found to be 0.016 cm (29.24 cm) in 2011 and 2.58 cm (26.65 cm) in 2012. The spatial distribution of monthly average pan-Antarctic snow freeboard found using this method is similar to what was observed from NASA's Ice, Cloud, and land Elevation Satellite (ICESat), where the difference (standard deviation) between October 2011-2017 CryoSat-2 mean snow freeboard and spring 2003-2007 mean freeboard from ICESat is 1.92 cm (9.23 cm). While our results suggest that this physical model and waveform fitting method can be used to retrieve snow freeboard from CryoSat-2, allowing for the potential to join laser and radar altimetry data records in the Antarctic, larger (~30 cm) regional differences from ICESat and along-track differences from ATM do exist, suggesting the need for future improvements to the method. Snow-ice interface elevation retrieval is also explored as a potential to obtain snow depth measurements. However, it is found that this retrieval method often tracks a strong scattering layer within the snow layer instead of the actual snow-ice interface, leading to an overestimation of ice freeboard and an underestimation of snow depth in much of the Southern Ocean but with promising results in areas such as the East Antarctic sector.

## 1 Introduction

Antarctic sea ice plays a complex yet important role in the earth system processes of the Southern Hemisphere. As the ice extent grows and shrinks over the course of a year, it can influence atmospheric circulations and temperatures (Cavalieri and Parkinson, 1981; Comiso et al., 2017), modify vertical and horizontal salinity profiles in the southern ocean (Aagaard and Carmack, 1989; Haumann et al., 2016), and even affect the biota of the south polar latitudes (Garrison, 1991; Legendre et al., 1992; Meiners et al., 2017). Perhaps most notably, the high albedo of snow-covered Antarctic sea ice means it reflects roughly 80 % of the incoming solar radiation back to space (Allison et al., 1993; Massom et al. 2001; Brandt et al., 2005; Zatko and Warren, 2015), helping to regulate the temperature of the south polar region and moderate the earth's energy budget. Unlike the Arctic Ocean, the Southern Ocean is unbounded by continents, resulting in geographically unlimited sea ice growth and vast areal extent. The average maximum extent of Antarctic sea ice is about 18.5 million km$^2$, occurring in September each year (Parkinson and Cavalieri, 2012). Despite a loss of sea ice extent in the Arctic since the late 1970's (Cavalieri and Parkinson, 2012), passive satellite remote sensing records of Antarctic sea ice have

shown a slight increase in areal extent over the same period at a rate of about 17,100 km$^2$ yr$^{-1}$ (Parkinson and Cavalieri, 2012). Over the past few years, passive satellite observations have shown considerable variability in Antarctic sea ice extent. A record maximum extent of 19.58 million km$^2$ was reached on 30 September 2013 (Reid et al., 2015), only to be topped in September 2014 when the extent reached 20.11 million km$^2$ (Comiso et al., 2017). Less than three years later, in March 2017, sea ice cover in the Antarctic dropped to just 2 million km$^2$, a record low in the satellite era (Turner and Comiso, 2017). This minimum followed an unparalleled retreat of Antarctic sea ice cover in 2016 (Turner et al., 2017).

In addition to ice extent, sea ice thickness is important for gauging the state of sea ice in the polar regions. Beginning in the mid-to-late 20$^{th}$ century, ship-based in situ measurements provided the only thickness data available in the Southern Ocean (Worby et al., 2008). More recently, satellite altimetry instruments and techniques have proven valuable in collecting sea ice thickness information. In order to calculate thickness from altimetry, the freeboard must first be computed. Freeboard is defined as either the height of the air-snow interface above the sea surface, termed the "snow freeboard" or "total freeboard", or as the height of the snow-ice interface above the sea surface, known as the "ice freeboard". Both types of freeboard can be used to compute thickness. Typically, altimeter-based sea ice thickness is derived by assuming a hydrostatic balance and combining the freeboard measurements with a measure of the snow depth atop sea ice as well as approximations for the densities of the snow, sea ice, and sea water. In the Antarctic, snow freeboard is used most often in this calculation (Li et al., 2018), which is usually obtained using measurements from a laser altimeter.

Zwally et al. (2008) made the first estimates of satellite laser altimeter-based Antarctic sea ice thickness by utilizing data from NASA's Ice, Cloud, and Land Elevation Satellite (ICESat) taken over the Weddell Sea. They computed the snow freeboard and combined it with snow depth data taken from the Advanced Microwave Scanning Radiometer for Earth Observing System (AMSR-E). After Zwally et al. (2008), several studies retrieved pan-Antarctic sea ice thickness from ICESat, each using slightly different methods. Kurtz and Markus (2012) combined ICESat freeboards with in situ density measurements and made the "zero ice freeboard" assumption that the snow depth was equal to the snow freeboard, and thus no independent snow depth measurements were required. Kern et al. (2016) compared multiple methods of computing thickness using ICESat freeboard data, by calculating snow depths from both AMSR-E and a static but seasonally-varying snow depth to thickness ratio. A new one-layer method was developed by Li et al. (2018) to compute thickness using ICESat data that built on the static ratio used by Kern et al. (2016) and incorporated a dynamic snow depth to thickness ratio for every data point. As these studies show, a large limitation to calculating Antarctic sea ice thickness from laser altimetry, regardless of the method used, is the uncertainty in the snow depth distribution on sea ice.

In addition to using laser altimetry to calculate sea ice thickness in the Antarctic, radar altimetry has also been used in recent years. Most radar altimeters operate in the Ku band at around 13.6 GHz, a frequency that has been shown to produce a dominant backscatter from the snow-ice interface (Beaven et al., 1995). The retrieved freeboard from radar altimetry, therefore, is generally assumed to be the ice freeboard especially when the snow is relatively dry and thin. Ku-band retrievals of ice freeboard have been employed in the Arctic (Laxon et al., 2003; Laxon et al. 2013; Giles et al., 2008), where the thinner and drier snow conditions tend to exist (Webster et al., 2018). In the Antarctic, radar freeboard calculations (and subsequent thickness calculations) are complicated substantially by the depth and variable vertical structure of the snow on top of the sea ice (Willatt et al. 2010; Price et al., 2015; Kwok, 2014). Due to the wealth of available moisture from the surrounding ocean, Antarctic sea ice experiences more frequent precipitation – and therefore greater snow depths – than that of the Arctic (Massom et al., 2001; Maksym et al., 2012). The deep snow can be heavy enough to depress the sea ice surface down near or even below the sea surface, leading to flooding and wicking of the seawater within the snowpack (Massom et al., 2001; Willatt et al., 2010) that can act to obscure returns from radar altimeters. Additionally, dense,

warm and/or moist snow can cause the dominant scattering surface to be located within the snowpack at a level that is higher than the snow-ice interface (Giles et al., 2008; Willatt et al., 2010; Willatt et al., 2011).

Freeboard retrievals that neglect range corrections for radar propagation through a snow layerare referred to as "radar freeboards". Radar freeboard was calculated in the Antarctic by Schwegmann et al. (2016), who used data from CryoSat-2 and Envisat to retrieve freeboard with the eventual aim to create a joined Envisat-CryoSat-2 sea ice thickness record. To counteract the effects of the snow layer on electromagnetic wave propagation, Paul et al. (2018) included a snow layer range correction to radar freeboards computed using CryoSat-2 and Envisat to retrieve ice freeboard over both Arctic and Antarctic sea ice. While the method put forth by Paul et al. (2018) demonstrates usefulness in reconciling thickness between Envisat and CryoSat-2, there still exist uncertainties in the sea ice thickness retrievals brought on by the validity of the snow depth climatology used in the corrections.

When using Ku-band altimetry for retrievals of freeboard and thickness, the largest source of uncertainty comes from the snow on sea ice. Uncertainty in the depth, salinity, and vertical structure can impact ranging and freeboard calculation (Armitage and Ridout, 2015; Ricker et al. 2015; Nandan et al. 2017). In order to counteract this uncertainty and improve the knowledge of the scattering effects of a snow layer on sea ice, our work aims to utilize Ku-band altimetry from CryoSat-2 to retrieve the elevation of the air-snow interface and subsequently the snow freeboard. While it is true that Ku-band radar pulses generally penetrate the snow surface on sea ice and have a dominant scattering layer beneath, what is often not included in freeboard retrieval algorithms, especially those depending on an empirical waveform evaluation, is the fact that there are physical and dielectric differences between air and snow (Hallikainen and Winebrenner, 1992; Stiles and Ulaby, 1980) that results in scattering – albeit comparatively weaker – from the air-snow interface (discussed in Sect. 3). Though this scattering is not typically the dominant return from radar pulses, it has been shown that it can be detected from airborne as well as ground-based sensors (Kurtz et al., 2013; Willatt et al., 2010). Satellite radar returns of the air-snow interface elevation would be important in the Antarctic where snow-ice interface returns are complex and uncertain, and provide the possibility for snow depth estimations from radar altimetry. Knowledge of the snow depth in the Antarctic would enable more accurate sea ice thickness calculations, given that recent studies of Antarctic sea ice thickness rely on passive microwave snow depth data (Kern et al., 2016), assumptions of snow depth being equal to snow freeboard (Kurtz and Markus, 2012), parameterizations of snow depth from both snow freeboard (Li et al., 2018) and multi-year ice fraction (Hendricks et al., 2018), or even treatment of the snow and ice layers as a single layer with a modified density (Kern et al., 2016).

Typically, CryoSat-2 pulses are limited by the receive bandwidth (320 MHz, corresponding to a vertical resolution of 0.234 m) and therefore not able to resolve the air-snow interface explicitly (Kwok, 2014). We show that through the use of a two-layer physical model that accounts for the scattering effects of a snow layer on top of sea ice, we are able to retrack the air-snow interface from CryoSat-2 radar waveforms, compute the surface elevation, and calculate snow freeboard. Our two-layer model builds on the single-layer method developed in Kurtz et al. (2014). This study begins by explaining the datasets that are used (Sect. 2), discusses the physical rationale (Sect. 3) and method (Sect. 4) of retrieving snow freeboard from CryoSat-2, and shows an initial validation of the approach (Sect. 5). Then, the freeboard calculation, results, and comparisons are discussed in Sect. 6. Finally, a discussion on the application to snow depth retrievals and possibility for future work is provided in Sect. 7 and Sect. 8.

## 2 Data sets

Data for this study primarily come from ESA's CryoSat-2 satellite, launched in 2010. The principle payload aboard CryoSat-2 is SIRAL, a Synthetic Aperture Interferometric Radar Altimeter, which has a frequency in the Ku-band at 13.575 GHz and a receive bandwidth of 320 MHz (Wingham et al., 2006). SIRAL operates in one of three modes: "low resolution" mode (LRM), "synthetic aperture" (SAR) mode, and "synthetic aperture interferometric" (SARin) mode. In the Southern Hemisphere, LRM is used over the

Antarctic continent and areas of open ocean and therefore is not considered in this study (Wingham et al., 2006). SAR and SARin data, which are taken over the sea ice zone and the Antarctic coastal regions, respectively, are both utilized in this work. Specifically, level 1B data from both of these operating modes are used. SAR level 1B data consist of 256 samples per echo while SARin data contain 512 samples per echo (Wingham et al., 2006). In order to maintain consistency between the two modes, both SAR and SARin data are here truncated to 128 samples per echo.

CryoSat-2 level 1B data utilize "multi-looking" to provide an average echo waveform for each point along the ground track. These multi-looked echoes correspond to an approximate footprint of 380 m along track and 1.5 km across track (Wingham et al., 2006). Within the level 1B data, the one-way travel time from the center range gate to the satellite center of mass is provided. This information is used to retrieve elevation above the WGS84 ellipsoid. To do so, we first multiply the one-way travel time by the speed of light in a vacuum. Then, geophysical and retracking corrections are applied following Kurtz et al. (2014). Geophysical corrections are applied by using the CryoSat-2 data products, which include the ionospheric delay, dry and wet tropospheric delay, oscillator drift, dynamic atmosphere correction (which includes the inverse barometer effect), pole tide, load tide, solid Earth tide, ocean equilibrium tide, and long period ocean tide. The retracking corrections are obtained through the waveform fitting method, discussed in Sect. 4. Adding the corrections to the raw range provides the surface elevation.

For this work, CryoSat-2 data from October 2011-2017 are utilized. October was chosen so that a substantial sea ice extent is present in each year of data and also so there is overlap with the spring ICESat campaigns, which ran roughly from October to November 2003-2009. Seven years of data allows for a longer-term average to be computed and facilitates better comparison with the ICESat spring seasonal average (Sect. 6.2).

Data from NASA's Operation IceBridge airborne campaign are used in multiple capacities throughout this study. First, IceBridge 2-8 GHz Snow Radar (Leuschen, 2014) and 13-17 GHz Ku-band radar altimeter (Leuschen et al., 2014) data are used to confirm the presence of scattering of the radar beam from the air-snow interface (Sect. 3). These data are taken from flights over the Weddell Sea on 13 October 2011 and 7 November 2012, which correspond to planned underflights of a contemporaneous CryoSat-2 orbit. This flight line is known as the "Sea Ice – Endurance" mission and is shown in Fig. 1. Second, these coincident observations are used in Section 5 for direct comparisons of elevations found between IceBridge and CryoSat-2, in order to validate this CryoSat-2 algorithm. Specifically, Airborne Topographic Mapper (ATM) elevation data (Studinger, 2014) are used and compared against that of CryoSat-2.

Sea ice freeboard data taken from ICESat between 2003-2007 (Kurtz and Markus, 2012) are used primarily as a comparative measure in this work. This product is gridded to 25 km and uses a distance weighted Gaussian function to fill gaps in the gridded data. Specifically, seasonal average freeboard values from the various ICESat campaigns are compared with CryoSat-2 monthly average freeboard data obtained using this algorithm. The austral spring ICESat freeboard dataset consists of measurements made from October and November 2003-2007 (Fig. 2). These ICESat freeboard and thickness data are publicly available online at neptune.gsfc.nasa.gov/csb/index.php?section=272.

Lastly, sea ice concentration data are used to filter out grid boxes that are largely uncovered with ice. We utilize the Comiso Bootstrap monthly average product, version 3, that provides sea ice concentration on a 25 km polar stereographic grid, and remove grid boxes with monthly average concentrations less than 50 %. This product is derived using brightness temperatures from Nimbus-7 SMMR and DMSP SSM/I-SSMIS passive microwave data (Comiso, 2017).

## 3 Observed Ku-band scattering of radar from Antarctic sea ice

While more recent studies have shown the effects that a snow layer can have on Ku-band ranging and freeboard retrievals (Armitage and Ridout, 2015; Ricker et al. 2015; Nandan et al. 2017),  past works that utilize Ku-band altimetry for ice freeboard retrieval tend to neglect scattering that occurs from the snow surface and volume, and assume that the dominant return occurs from the snow-ice interface (Beaven et at., 1995; Laxon et al. 2013; Kurtz et al. 2014). For most cases, especially in the Arctic where the snow cover is relatively thin and dry, this assumption is generally valid (Willatt et al., 2011; Armitage and Ridout, 2015). However, the physical differences between air and snow indicate that scattering can occur from the air-snow interface as well (Hallikainen and Winebrenner, 1992). This air-snow interface scattering is the fundamental basis for measuring snow freeboard using radar altimetry. Kwok (2014) used Operation IceBridge data to find that scattering from the air-snow interface does contribute to the return at Ku-band frequencies. To further prove this fact, we use Operation IceBridge echogram data from the Ku-band and snow radars (Fig. 3) that provide a vertical profile of the radar backscatter along the flight path displayed in Fig. 1. These echograms come from the November 2012 campaign. Comparing the lower-frequency snow radar, which is known to detect the air-snow interface, with the higher frequency Ku-band radar altimeter, one can see the difference in scattering between the snow-covered floe points and the leads in both radar profiles.

In this study, a simple "peak picking" algorithm is employed to mark the vertical locations of both the maximum backscatter and the first point that rises 10 dB above the noise level for each horizontal point along the flight line. While not explicitly extracting layers from the IceBridge data, these points are used as initial guesses of the air-snow and snow-ice interfaces into the model (Sect. 4). These initial guesses are not exactly the expected backscatter coefficients from the two layers, but instead a rough approximation from their peak powers. The peak-picked air-snow interface power is compared to that of the maximum (assumed snow-ice interface) power, as displayed in Fig. 4. This frequency distribution shows that for the 2012 IceBridge campaign over Antarctic sea ice, the difference of the air-snow interface power from the maximum power is smaller for the snow radar, with a mean of 12.94 dB, than for the Ku-band altimeter, which has a mean difference of 14.00 dB. This result is expected, as it means that the scattering power from the air-snow interface is closer in magnitude to that of the snow-ice interface in snow radar returns. However, the curves have a similar distribution and mean, indicating that the Ku-band radar return likely consists of scattering from the air-snow interface as well. Overall, a comparison of the IceBridge radars provides further evidence that scattering of Ku-band radar pulses can occur at the air-snow interface. The following sections utilize this notion to retrieve snow freeboard from CryoSat-2 returns.

## 4 Surface elevation retrieval methodology

In this section, we introduce a new two-layer retrieval method that expands on the single layer method employed by Kurtz et al. (2014). Following that work, this study retrieves surface elevation from CryoSat-2 data by first using a physical model to simulate return waveforms from sea ice. Then, a least-squares fitting routine is used to fit the simulated waveform to the CryoSat-2 level 1B data. Sea ice parameters, including the surface elevation, can then be computed from the fit waveform. The following section describes this process. For a more detailed derivation of the theoretical basis surrounding the physical model and waveform fitting routine, see Kurtz et al. (2014).

### 4.1 Physical waveform model

When assuming a uniformly backscattering surface, Kurtz et al. (2014) expressed the received radar echo, $\Psi(\tau)$, as

$$\Psi(\tau) = P_t(\tau) \otimes I(\tau) \otimes p(\tau) \tag{1}$$

where $\tau$ is the echo delay time relative to the time of scattering from the mean scattering surface and $\otimes$ represents a convolution of the compressed transmit pulse, $P_t(\tau)$, the rough surface impulse response, $I(\tau)$, and the surface height probability density function, $p(\tau)$ (Brown, 1977; Kurtz et al., 2014). The terms are defined as

$$P_t(\tau) = p_0 \text{sinc}^2(B_w \tau), \tag{2}$$

where $p_0$ is the peak power of the pulse and $B_w$ is the received bandwidth,

$$p(\tau) = \frac{1}{\sqrt{2\pi}\sigma_c} exp(-\frac{1}{2}(\frac{\tau}{\sigma_c})^2), \tag{3}$$

where c is the speed of light in vacuo and $\sigma_c = \frac{2\sigma}{c}$, the standard deviation of the surface height in the time domain, and

$$I(\tau) = \frac{\lambda^2 G_0^2 D_0 c \sigma^0(0\degree)}{32\pi h^3 \eta} \sum_{k=-\frac{N_b-1}{2}}^{\frac{N_b-1}{2}} H\left(\tau + \frac{\eta h \xi_k^2}{c}\right) exp\left[\frac{-2\xi_k^2}{\eta^2}\left(\frac{1}{\gamma_1^2}+\frac{1}{\gamma_2^2}\right) + \frac{c\eta}{h\gamma_1^2}\left(\tau+\frac{\eta h \xi_k^2}{c}\right)\right] \int_{h\gamma_1^2}^{c\eta} d\theta \, exp\left[-4\xi_k\sqrt{\frac{c}{h\eta^3}\left(\tau+\frac{\eta h \xi_k^2}{c}\right)}\cos\theta\left(\frac{1}{\gamma_1^2}+\frac{1}{\gamma_2^2}\right) - \right.$$

$$\left. \frac{2c\cos(2\theta)}{h\eta\gamma_2^2}\left(\tau+\frac{\eta h \xi_k^2}{c}\right)\right]\left(1+\frac{\alpha}{h^2}\left(\left(\frac{h\xi_k}{\eta}\right)+\frac{ch}{\eta}\left(\tau+\frac{\eta h \xi_k^2}{c}\right)+2\left(\frac{h\xi_k}{\eta}\right)\cos\theta\sqrt{\frac{ch}{\eta}\left(\tau+\frac{\eta h \xi_k^2}{c}\right)}\right)\right)^{-3/2}\left(\sum_{n=0}^{N_b}\left(0.54-0.46\cos\left(\frac{2\pi n}{N_b}-\pi\right)\right)\cos\left(2k_0 v_s\left(n-\right.\right.\right.$$

$$\left.\left.\left.\frac{N_b}{2}\right)\sqrt{\frac{c\tau}{\eta h}+\xi_k^2}\cos\theta - \xi_k\right)\right)^2, \tag{4}$$

where the variables (average values , when applicable, following Kurtz et al. (2014)) for CryoSat-2 are as follows: $\lambda$ (0.0221 m) is the center wavelength, $G_0$ (42 dB) is the one-way antenna gain, $D_0$ (30.6 dB) is the one-way gain of the synthetic beam, c (299792485 m s$^{-1}$) is the speed of light in vacuo, $\sigma^0(0\degree)$ is the nadir backscatter coefficient, h (725 km) is the satellite altitude, $\eta$ (1.113) is a geometric factor, $N_b$ (64) is the number of synthetic beams, $\tau$ is the echo delay time, $\xi_k$ is the look angle of the synthetic beam k from nadir, H is a Heaviside step function, $\gamma_1$ (6767.6) is the elliptical antenna pattern term 1, $\gamma_2$ (664.06) is the elliptical antenna pattern term 2, $\alpha$ is the angular backscattering efficiency, $k_0$ (284.307 m$^{-1}$) is the carrier wave number, $v_s$ (7435 m s$^{-1}$) is the satellite velocity, $\sigma$ is the standard deviation of surface height, and $B_w$ (320 MHz) is the received bandwidth.

Under the assumption that only surface scattering is present and occurs from the snow-ice interface alone (i.e. no surface scattering from the air-snow interface nor volume scattering from within the snow or ice layers), Eq. (1) is able to accurately model a received CryoSat-2 echo over the Arctic (Kurtz et al., 2014). However, due to thicker snow depths on Antarctic sea ice as compared to the Arctic, scattering effects from the snow layer cannot be neglected when retrieving surface elevation. Therefore, Eq. (1) is here modified to become

$$\Psi(\tau) = P_t(\tau) \otimes I(\tau) \otimes p(\tau) \otimes v(\tau) \tag{5}$$

where $v(\tau)$ is the scattering cross section per unit volume as a function of echo delay time (Kurtz et al., 2014). Following Arthern et al. (2001) and Kurtz et al. (2014), $v(\tau)$ is defined in terms of physical parameters including the surface backscatter coefficients of snow and ice, $\sigma^0_{surf-snow}$ and $\sigma^0_{surf-ice}$, respectively, and the integrated volume backscatter of snow and ice, $\sigma^0_{vol-snow}$ and $\sigma^0_{vol-ice}$, respectively. Together, the total backscatter can be written as

$$\sigma^0 = \sigma^0_{surf-snow} + \sigma^0_{vol-snow} + \sigma^0_{surf-ice} + \sigma^0_{vol-ice}. \tag{6}$$

For snow on sea ice, $v(\tau)$ becomes

$$v(\tau) = \begin{cases} 0, \tau < -\frac{2h_s}{c_{snow}} \\ \sigma^0_{surf-snow}\delta\left(\tau+\frac{2h_s}{c_{snow}}\right)+\sigma^0_{vol-snow}k_{e-snow}exp\left[-c_{snow}k_{e-snow}\left(\tau+\frac{2h_s}{c_{snow}}\right)\right], 0 < \tau \le -\frac{2h_s}{c_{snow}} \\ \sigma^0_{surf-ice}k_{t-snow}^2 exp\left[-\frac{k_{e-snow}h_s}{2}\right]\delta(\tau)+\sigma^0_{vol-ice}k_{e-ice}exp\left[-\frac{k_{e-snow}h_s}{2}-c_{ice}k_{e-ice}\tau\right], \tau \ge 0 \end{cases}, \tag{7}$$

which accounts for signal attenuation in the snow and ice layers and loss of power at the air-snow and snow-ice interfaces. Eq. (7) comes from Kurtz et al. (2014) and uses the form of $\tau = 0$ at the snow-ice interface. In Eq. (7),

$$\sigma^0_{vol-snow} = \frac{\sigma_{vol-snow}k^2_{t-snow}}{k_{e-snow}},$$ (8)

$$\sigma^0_{vol-ice} = \frac{\sigma_{vol-ice}k^2_{t-snow}k^2_{t-ice}}{k_{e-ice}}.$$ (9)

Static parameters in Eqns. (7) – (9) are given values to model a snow layer on sea ice. We assign the two-way extinction coefficients of snow, $k_{e-snow}$, and sea ice, $k_{e-ice}$, to be 0.1 m$^{-1}$ and 5 m$^{-1}$, respectively, following Ulaby et al. (1986). The speed of light through snow and ice are $c_{snow}$ and $c_{ice}$, respectively, where $c_{snow} = \frac{c}{n_{snow}}$ and $c_{ice} = \frac{c}{n_{ice}}$. Here, $n_{snow}$ = 1.281 and $n_{ice}$ = 1.732, where $n_{snow}$ corresponds to a snow layer with a density of 320 kg m$^{-3}$ (Tiuri et al., 1984, Ulaby et al., 1986). A density of 320 kg m$^{-3}$ was chosen as an assumption to best represent pan-Antarctic snow on sea ice following results from several in situ surveys (Massom et al., 2001; Willatt et al., 2010; Lewis et al., 2011). Finally, $k_{t-snow}$ and $k_{t-ice}$ are the transmission coefficients between the air-snow and snow-ice interfaces, respectively. Both transmission coefficients are generally close to one (Onstott, 1992); we use values of $k_{t-snow}$=0.9849 and $k_{t-ice}$ = 0.9775 as calculated from the Fresnel reflection coefficient using the values of $n_{snow}$ and $n_{ice}$. $h_s$, the snow depth, is computed from the echo delay shift of the air-snow and snow-ice interfaces, free parameters $t_{snow}$ and t respectively, which are discussed in Sect. 4.3. The remaining free parameters are given as inputs to the model and are defined in the following section.

The main assumption in this approach is that scattering is expected to come from two defined layers (i.e. the air-snow and snow-ice interfaces) and uniformly throughout the volume. Antarctic sea ice can exhibit complex layer structures that could obscure this simple two-layer method, however, no pan-Antarctic understanding of snow-covered sea ice composition currently exists. Therefore, this two-layer assumption is utilized as an approximation of the broad-scale sea ice cover.

**4.2 Waveform fitting routine**

To fit the modelled waveform to CryoSat-2 data, a bounded trust region Newton least-squares fitting routine (MATLAB function *lsqcurvefit*) is employed. This routine fits the model to the data by iteratively adjusting model input parameters and calculating the difference between the modelled and CryoSat-2 level 1B waveform data, until a minimum solution – or the established maximum number of iterations – is reached. Building off of Kurtz et al. (2014), this process can be shown with the equations

$$P_m(\tau) = A_f L(\tau, \alpha, \sigma) \otimes p(\tau, \sigma) \otimes \nu\left(\tau, t_{snow}, \sigma^0_{surf-snow}, \sigma^0_{surf-ice}, \sigma^0_{vol-snow}, \sigma^0_{vol-ice}\right)$$ (10)

and

$$min \sum_{i=1}^{128}[P_m(\tau_i) - P_r(\tau_i + t)]^2,$$ (11)

where L is a lookup table of $P_t(\tau) \otimes I(\tau)$ as defined in Kurtz et al. (2014), $P_m$ is the modelled waveform, $P_r$ is the observed echo waveform, and $\tau_i$ is the observed echo power at point *i* on the waveform. These equations result in nine free parameters: the amplitude scale factor, $A_f$, the echo delay shift factor at the air-snow and snow-ice interfaces, respectively $t_{snow}$ and t, the angular backscattering efficiency, $\alpha$, the standard deviation of surface height, $\sigma$, and the terms that together make up the total backscatter, $\sigma^0_{surf-snow}$, $\sigma^0_{surf-ice}$, $\sigma^0_{vol-snow}$, and $\sigma^0_{vol-ice}$. These parameters are adjusted with each iteration of the fitting routine and are explained further in Sect. 4.3.1 and Sect. 4.3.2. An initial guess for each of the free parameters – in addition to upper and lower bounds – is provided to the fitting routine. Doing so ensures that the solution reached will closely resemble that of the physical system. Approaches for determining the initial guesses for both lead and floe characterized echoes are outlined in the following section.

This algorithm uses the squared norm of the residual ("resnorm") as a metric for goodness of fit. Modelled waveforms with a resnorm less than or equal to 0.3 are considered to be good fits and have the output parameters used in the retracking correction calculation and surface elevation retrieval. Waveforms with greater fitting error are run again using a different initial guess for $\alpha$. If the resnorm is still high, the CryoSat-2 echo is not used in the retrieval process. Figure 5 shows a spatial distribution of the mean October resnorm values for 2011-2017. The largest residuals are consistently located around the ice edge and near to the continent, while the smallest are collocated with areas of high lead-type fraction (Fig. 5), such as the Ross Sea. Since the specular lead waveforms are easily fit with little residual, the overall average distribution shown here is consistently under 0.3 (total mean of 0.13). However, many floe-type points have values closer to the 0.3 threshold. Although we have observed that a resnorm threshold of 0.3 results in reasonably representative modelled waveforms, we understand that the use of a single metric can oversimplify the goodness of fit and leaves room for errors in the shape of the modelled waveform. Future work will look into a incorporating a more comprehensive metric for goodness of fit.

### 4.3 Lead / floe classification

Prior to constructing a physical model and fitting it to the data, each CryoSat-2 echo is first characterized as either a lead or a floe based on parameters derived from the individual waveform. Specifically, the pulse peakiness (PP) and stack standard deviation (SSD) parameters are used to distinguish between the two surface types, following Laxon et al. (2013). PP is defined as

$$PP = max(P_r) \sum_{i=1}^{128} \frac{1}{P_{r(i)}} \tag{12}$$

from Armitage and Davidson (2014). SSD comes from the CryoSat-2 level 1B data product and is due to the variation in the backscatter as a function of incidence angle (Wingham et al., 2006). Figure 5 shows average detection rates for lead and floe points using this method, discussed in the following sections.

### 4.3.1 Leads

CryoSat-2 echoes are categorized as leads if the return waveform has a PP > 0.18 and a SSD < 4 (Laxon et al., 2013). Since by definition leads have no snow cover, it is assumed that all scattering of the radar pulse originates from one surface. In this case, that surface is either refrozen new ice or open water. It is also assumed that no volume scattering occurs from leads. Therefore, the volume scattering term in Eq. (10) goes to a delta function at $\tau = 0$, resulting in four free parameters: the amplitude scale factor, $A_f$, the echo delay shift factor, t, the angular backscattering efficiency, $\alpha$, and the standard deviation of surface height, $\sigma$. The initial guess for $A_f$ is set equal to the waveform peak power, with the bounds set to +/- 50 % of the peak power. The echo delay shift, t, is given an initial guess equal to the point of maximum power, denoted with $t_i$. $\sigma$ is first estimated to be 0.01 for lead points, with bounds taken to be $0 \leq \sigma \leq 0.05$. The initial guess for $\alpha$, denoted as $\alpha_0$, is calculated as the ratio of tail to peak power and uses a mean of the 10ns following the location of peak power. The bounds of $\alpha$ are $\frac{\alpha_0}{100} \leq \alpha_0 \leq 100\alpha_0$. Using the above initial guesses in the fitting routine leads to a modelled waveform that well represents the CryoSat-2 data over leads (Kurtz et al., 2014). The echo delay factor, t, provides the location of the surface as a function of radar return time, which is used in the surface elevation retrieval of each lead-classified echo. The largest fraction of lead-classified points occurs in the Ross Sea, consistent with the location of the Ross Sea Polynya (Fig. 5). However, it is also a region known for new-ice formation that could return specular lead-type waveforms, and potentially lead to an overestimation of the sea surface height (discussed in section 6).

## 4.3.2 Floes

Radar echoes with a PP < 0.09 and a SSD > 4 are classified as sea ice floes (Laxon et al., 2013). Due to the presence of a snow layer on top of the sea ice, all nine free parameters (introduced in Sect. 4.2) are employed. These include the four mentioned in the previous section, as well as $t_{snow}$, the echo delay shift factor of the air-snow interface, $\sigma_{surf-snow}^0$, $\sigma_{vol-snow}^0$, $\sigma_{surf-ice}^0$, and $\sigma_{vol-ice}^0$. The initial guess and bounds for $A_f$ is taken to be the same as used for lead points, while the remaining 8 differ from leads. For $t_{snow}$, the initial guess ($t_{i-snow}$) comes from the ICESat datasets of the seasonal average total freeboard. We use the "zero ice freeboard" assumption (Kurtz et al., 2012) that the snow-ice interface is depressed to the sea surface, meaning the ICESat freeboard would be approximately equal to the snow depth. Though this assumption is generally thought to be valid in the Antarctic, it may not hold true in all regions of the Antarctic (Adolphs, 1998; Weissling and Ackley, 2011; Xie et al., 2011; Kwok and Maksym, 2014). Therefore, this fitting routine attempts to adapt and move away from the zero ice freeboard assumption, with the results being explored in later sections. The ICESat freeboard height at the location of each CryoSat-2 radar pulse is taken and converted in terms of radar return time, which provides a suitable initial guess of the air-snow interface. Bounds of $t_{i-snow}$ are taken to be +/- 5 ns. The initial guess for t ($t_i$) is taken to be the first point where the waveform power reaches 70 % of the power of the first peak, following Laxon et al. (2013). This is a commonly used threshold retracking method to detect the snow-ice interface from CryoSat-2. Bounds are taken to be +/- 6 ns. $\sigma$ is first estimated to be 0.15 for floe points, with bounds set to $0 \leq \sigma \leq 1$. The initial guess for $\alpha$ is similar to that in the lead characterization, with the exception that the mean power of points between 90 ns and 120 ns is used in the ratio of tail to peak power. Bounds for $\alpha_0$ are set as $\frac{\alpha_0}{100} \leq \alpha_0 \leq 100\alpha_0$.

The remaining surface backscatter coefficients and integrated volume backscatter of snow and ice are initially estimated using values taken from Operation IceBridge Ku-band radar echograms from the Weddell Sea flights. Estimation of the surface backscatter comes from an average of all valid peaks chosen from the echogram peak-picker for the air-snow and snow-ice interfaces of both flights. The snow and ice volume backscatter values are parameterized using average layer backscatter values between the two interfaces and 10 range bins beyond the snow-ice interface, respectively. The initial guesses (bounds) are set to be as follows: $\sigma_{surf-snow}^0$ = -15 dB (+/- 5 dB), $\sigma_{vol-snow}^0$ = -11 dB (+/- 5 dB), $\sigma_{surf-ice}^0$ = -1 dB (+/- 10 dB), and $\sigma_{vol-ice}^0$ = -8 dB (+/- 10 dB). The largest fraction of floe-type points are found in the Weddell Sea and along the ice edge, where older and rougher ice is generally found (Fig. 5). These distributions compare qualitatively to that found in Paul et al. (2018), with the exception that this method finds a larger region of lead-type dominant waveforms in the Ross Sea than Paul et al. (2018).

The modelled waveform (examples shown in Fig. 6) is sensitive to the initial guess provided, and therefore care was taken to ensure the initial guesses come from physically realistic values. A change in the initial guesses results in different final fits, and subsequently a different freeboard distribution. Figure 6 shows a waveform sensitivity study looking at a variety of modeled waveforms that differ only in the initial guess for the standard deviation of surface height ($\sigma$, top) and the total backscatter coefficient ($\sigma^0$, bottom). The range of $\sigma$ was taken to be between 0.01 (very smooth surface) to 0.4 (rough surface), while $\sigma^0$ was varied between three different parameterizations: values from Kurtz et al. (2014), values taken from the IceBridge Snow Radar data, and from the Ku-band data (above). The resulting freeboard distributions found using an initial guess of $\sigma$ = 0.35 and $\sigma^0$ taken from Kurtz et al. (2014) are shown as a difference from the values chosen in this study ($\sigma$ = 0.15, $\sigma^0$ taken from Ku-band radar data) in Fig. 6 (C and F). In this case, the effect of altering the backscatter parameterization had a larger effect on freeboard than altering $\sigma$. It is evident that physically inconsistent initial guesses can result in altered freeboard distributions, with the magnitude of the impact potentially being large (broad-scale difference of ~25 cm in Fig. 6 C). While this uncertainty surely adds to that of the overall results, the use of physically consistent first guesses acts to reduce the uncertainty as much as possible. Thus, a future area of study will be to determine better empirical first guess choices for the static free parameters currently used in the model.

## 5 Initial validation

To evaluate the performance of this algorithm, the returned surface elevation is compared to independent measurements of surface elevation from Operation IceBridge. Specifically, ATM data taken from the IceBridge underflight of the CryoSat-2 orbit (Fig. 1) is compared with retracked CryoSat-2 elevation data derived using this algorithm. The comparison is done between surface elevation measurements before any freeboard calculations are made, ensuring that differences observed are a factor of the retrieval alone. In order to facilitate a direct comparison, ATM level 2 Icessn elevation data are averaged to the same ground footprint size as a CryoSat-2 echo. Additionally, equivalent geophysical corrections are computed and applied (following Yi et al., 2018) to both the CryoSat-2 and ATM datasets, ensuring that both measurements are in the same frame of reference. These geophysical corrections include effects from tides, which are computed using the TPXO8-Atlas model (Egbert and Erofeeva, 2002), the mean sea-surface height, which are computed using the Technical University of Denmark DTU15MSS dataset (Anderson et al., 2016), and the dynamic atmosphere, which are computed using correction data from the Mog2G model (Carrère and Lyard, 2003).

Surface temperatures from MERRA-2 (GMAO, 2015) at the midpoint time of both IceBridge flights are found in Fig. 1. The 2011 flight had a large (about 20 °C) north-south temperature gradient that could result in different snow and ice properties along the flight line, and thus could explain differences observed along the line. In 2012, there was almost no temperature gradient along the flight line. Additionally, surface temperatures remained below freezing for the two weeks prior to both flights, with light snowfall of around 5 mm day$^{-1}$ occurring three (four) days prior to the flight in 2011 (2012) but stopping two (three) days before the flight. The time difference between the IceBridge flight and CryoSat-2 overpass was between 0 and 3.1 hours in 2011 and between 0 and 2.2 hours in 2012.

Figure 7 (A and B) shows ATM and CryoSat-2 surface elevation profiles from both the 2011 and 2012 IceBridge underflights. In these cases, the initial guess for the air-snow interface location in the CryoSat-2 fitting routine comes from the ICESat seasonal average dataset. Overall, the CryoSat-2 retracked elevation profiles capture the general trends found in the ATM profiles. The mean difference in elevation of CryoSat-2 from ATM for the entire flight line is 0.016 cm in 2011 and 2.58 cm in 2012. A frequency distribution of this difference is shown in Fig. 7 (C and D). Both years display a Gaussian-like distribution centered near zero (i.e. no difference) with standard deviations of 0.29 m in 2011 and 0.27 m in 2012. It is likely that some of the differences are due to initial temporal and spatial discrepancies between the IceBridge and CryoSat-2 data collections. Correlations coefficients are 0.44 in 2011 and 0.40 in 2012, which, although in the low-to-mid range, is likely brought on by the inherent noise in the data at the shot-to-shot level and non-overlapping footprints of the two sensors (Yi et al., 2018). Although mean resnorm values from the CryoSat-2 flight lines are 0.1124 in 2011 and 0.0990 in 2012, signifying good fits, it is still possible that errors in air-snow interface elevation could have arisen from errors in fits that were below the single-metric resnorm threshold but not representative of the actual CryoSat-2 waveform. This resnorm threshold is likely the cause of the "jumps" seen in the CryoSat-2 data, as testing a higher resnorm threshold led to more jumps, while a testing a lower resnorm threshold led to fewer jumps, but worse agreement to ATM. There also appears to be a slight underestimation of ATM by CryoSat-2 in both profiles, which could be brought on by the original footprint sizes, as the smaller ATM footprint is more sensitive to small-scale peaks/ridges than CryoSat-2.

Overall, this initial validation shows the potential of our CryoSat-2 algorithm to retrieve reasonable surface elevation measurements over Antarctic sea ice. This promising result warrants further exploration into freeboard retrieval using this method, discussed in the next section.

## 6 Snow freeboard retrieval

### 6.1 Freeboard calculation

The retrieved elevation of the air-snow interface from this method is used to calculate the snow freeboard of Antarctic sea ice. First, one month of CryoSat-2 data is processed at a time and the outlying data points are filtered out to reduce the inherent noise of the data. The filtering is done by removing any point that has an output parameter more than three standard deviations away from the mean of the respective parameter. These output parameters include quantities such as the surface elevation, retracking correction, PP, SSD, and $\tau$. Additionally, points with a $\tau$ value less than -100 ns were found to produce anomalous surface elevations and therefore are filtered out. Then, surface elevation data consisting only of echo points characterized as leads are gridded to a 25 km polar stereographic grid and averaged over the month. Grid boxes with fewer than five points and/or monthly concentrations less than 50 % are ignored. This grid is effectively the mean sea surface elevation. Snow freeboard is calculated by taking each surface elevation point along the CryoSat-2 orbit and subtracting the corresponding mean sea surface elevation value. Any snow freeboard points less than -0.1m and greater than 2.1m are filtered out. Between the initial filtering and this freeboard filtering, 41.68% of the total waveforms are filtered out, leaving 58.32% as valid waveforms. This process is done from the entire month of data, and the remaining freeboard values are gridded to 25 km to produce a map of the monthly mean snow freeboard. To study multi-year means for a given month, each monthly snow freeboard grid is averaged over a range of years. In this case, grid boxes with data from fewer than two years are ignored. Both the monthly and multi-year mean snow freeboard grids are smoothed by taking the average of all grid boxes within 2 grid boxes in all directions, which reduces the spatial resolution to 125 km. Smoothing is applied to reduce noise in the CryoSat-2 data and also to fill gaps in the data.

Figure 8 shows maps of October monthly averaged snow freeboard values from 2011-2017 as well as the mean of all seven years. The freeboard distribution corresponds well to what is expected in the Antarctic: the largest values occur in the Weddell and Amundsen seas – where ice production and heavy snow falls are typically prevalent – as well as along the coast of East Antarctica – where snowfall accumulation is also typically large. The smallest values tend to be found off the coast of East Antarctica between 0° and 90° E. Additionally, the region of low freeboard shown in the Ross Sea each year is consistent with the presence of young ice from the Ross Sea Polynya, but could be biased lower due to the large region of lead-type waveforms classified in the area, leading to a higher sea surface height and lower freeboard. While the overall pattern remains similar in each map, there is clear inter-annual variability. For example, the Amundsen Sea region along the Antarctic coastline exhibits a widespread area of very large (over 50 cm) freeboard in 2011, while the same coastal region between 100° W and 150° W shows values between 20 and 35 cm in 2016. Thicker snow freeboard can be found adjacent to the ice extent edge in each of the years, with the average map clearly showing greater freeboard values along the ice edge in the Western Pacific Ocean (about 90° E to 180° E). This thick freeboard at the ice edge is consistent with the older and thicker ice that has been previously found in the Antarctic frontal ice zone (Nghiem et al., 2016), but could also be due to surface waves penetrating the ice cover, resulting in an altered floe size distribution (Fox and Haskell, 2001) and also a high freeboard bias. Additionally, the high freeboard found here could be a product of the lower CryoSat-2 data density further from the pole as well as the variety of different ice types found in the frontal ice zone.

A time series of mean October snow freeboard from 2011 to 2017 found using this method is shown in Fig. 9, with total sea ice area plotted for reference (Fetterer et al., 2017). Apart from slight increases in freeboard from 2012 to 2013 and 2016 to 2017, there is an overall decrease found between 2011 and 2017 of 0.5 cm yr[-1]. The smallest measured freeboard occurred in 2016 (25.8 cm) which is collocated with a minimum in sea ice area that occurred in the same year. The total average snow freeboard in October from 2011 to 2017 is found to be 27.6 cm with a standard deviation of 13.0 cm. Interestingly, the sea ice area and snow freeboard time

series appear highly correlated between 2011 and 2017 (r=0.77) alluding to a potential relationship between freeboard/thickness and area in the Antarctic. This relationship, however, is beyond the scope of this paper and will be explored in future work.

## 6.2 Pan-Antarctic freeboard comparisons

To assess the performance of this algorithm on a pan-Antarctic scale, monthly averaged freeboard values from CryoSat-2 are compared with seasonal average freeboard from ICESat. Figure 10 shows a difference map between CryoSat-2 and ICESat total freeboard, where positive (negative) values indicate regions where CryoSat-2 measures greater (smaller) freeboard as compared to ICESat. The most notable difference occurs in the Weddell Sea off of the Antarctic Peninsula, where CryoSat-2 records a freeboard value much lower (around 30 cm) than ICESat. A similar region can be found in the Amundsen Sea, where CryoSat-2 measurements are again less than ICESat. CryoSat-2 measures a larger freeboard along most of the sea ice edge, as well as along the Antarctic coast from about 20° W to 60° E. Apart from these areas of noticeable differences between the two data sets, the remainder of the sea ice zone is fairly comparable among both. The total mean difference is only 2.9 cm with a standard deviation of just under 10 cm and a mode difference of 0.8 cm (Fig. 10).

Though this compatibility is encouraging, it is important to note that comparison is indirect in nature. The CryoSat-2 dataset covers October 2011-2017, seven years of data, while this ICESat dataset covers October-November 2003-2007, five years of data. These non-overlapping time periods have different lengths, and the ICESat dataset contains data from October and November in some of the campaigns. Therefore, this comparison shows that our algorithm can produce results similar to the average values found with ICESat, but requires temporally coincident data, such as that forthcoming from ICESat-2, to best assess the accuracy of the retrieval approach.

Qualitatively, the snow freeboard distribution found in Fig. 8 is comparable to that shown in Schwegmann et al. (2016) and Paul et al. (2018). In all studies, the largest freeboard is found along the coast in the Amundsen Sea, East Antarctica, and in the Weddell Sea, while the smallest freeboard is found in the Ross Sea and off East Antarctic between 0° and 90° E. Similar to what was found in the comparison with ICESat, both Schwegmann et al. (2016) and Paul et al. (2018) find a higher freeboard immediately off the Antarctic Peninsula near the Larson Ice Shelf than is found with this method, which could signal a regional difficulty to retrieve snow freeboard using this algorithm or a complication with the thicker and/or rougher ice that tends to be found in this region. However, these comparisons are still rather indirect, given that the prior works retrieve radar freeboard (Schwegmann et al., 2016) and ice freeboard (Paul et al., 2018) while this method retrieves snow freeboard. Once again, coincident measurements of snow freeboard from ICESat-2 will be invaluable as a comparative tool.

## 7 Application to snow depth retrievals

Given that this algorithm outputs the location of both the air-snow and snow-ice interfaces as a function of radar return time, it seems logical that snow depth could be extracted from these data. Likely, however, the complexities of Antarctic sea ice inhibit this method in tracking the correct snow-ice interface, resulting in a lower-than-expected (judging from passive microwave measurements (Markus and Cavalieri, 1998) and in-situ surveys (Massom et al., 2001)) snow depth distribution . Figure 11 shows a map of the average October 2011-2017 snow depth on sea ice, calculated by subtracting the snow-ice interface elevation from the air-snow interface elevation. It can be seen that for a majority of the Antarctic, a snow depth of around 0.1 m is present. This algorithm appears to be tracking the dominant sub-surface return as a layer within the snowpack as opposed to the ice interface itself, as has been seen in previous studies (e.g. Giles et al., 2008; Willatt et al., 2010). A potential explanation is that the complex snow stratigraphy found during in situ surveys of the Antarctic sea ice pack (Massom et al., 2001; Willatt et al., 2010; Lewis et al., 2011) and attenuation due to seawater flooding

and wicking could be prevalent throughout the Antarctic, and that layers of ice and/or brine could be responsible for an interface return that is higher than the actual snow-ice interface.

A similar result is found when comparing retrieved CryoSat-2 snow depths in the Weddell Sea to that from Operation IceBridge. Using the peak-picking algorithm on IceBridge data from the 13 October 2011 flight line, we calculate an approximate mean snow depth of 0.26 m. This value is close to the snow depth that was calculated by Kwok and Maksym (2014, table 2 S2-S4) for the same flight line (approximately 0.29 m). From CryoSat-2, the mean snow depth along the flight line is found to be 0.15 m, which is lower than the measured values potentially due to the much larger footprint size and more limited bandwidth from the satellite data.

Despite the widespread small snow depth values, the region off the coast of East Antarctica in Fig. 11 (between 90° E and 60° E) exhibits values closer to what is expected. Here, there is a greater snow depth of around 0.3 m. This region is known to have positive ice freeboard values (Worby et al., 1998; Maksym and Markus, 2008; Markus et al., 2011) meaning that flooding and saltwater intrusion would play less of a role than in other areas. The near-realistic snow depth measurements here provide evidence that our algorithm could be effective in retrieving snow depth under certain snow conditions, seasons, or locations, but speaks to the inherent complexity and uncertainty associated with Antarctic sea ice. Furthermore, the fact that these snow depth measurements are not higher over other areas of known positive ice freeboard, such as the western Weddell Sea, could signal issues regional issues of the algorithm to retrieve the snow-ice interface. More work is needed in evaluating the tracking of the snow-ice interface using this method to use it together with the air-snow interface for snow depth on sea ice estimation.

## 8 Conclusions and future work

In this work, a method for retrieving snow freeboard from CryoSat-2 data is developed. It is based off the fundamental idea that scattering of Ku-band radar pulses can originate from the air-snow interface of snow on sea ice. We incorporate this scattering into a physical waveform model and use a least squares fitting routine to fit the model to CryoSat-2 level 1B waveforms. The returned fit waveform and associated parameters includes, among others, the location of the air-snow interface as a function of radar return time. We are able to use that location to retrack the snow surface elevation, and from this, calculate snow freeboard. Through a comparison of this method with independent measurements, we are able to evaluate the performance of our retrieval. Specifically, surface elevation measurements from Operation IceBridge ATM, taken in October 2011 and November 2012 along a coincident flight line, help to provide an initial confirmation that the retrieval results were comparable to other data sources. Mean (standard deviation) elevation differences between ATM and CryoSat-2 were found to be just 0.016 cm (29.24 cm) in 2011 and 2.6 cm (26.65 cm) in 2012. Seasonal averaged freeboard data from ICESat allowed for the comparison of the pan-Antarctic freeboard. Though the CryoSat-2 and ICESat freeboard data come from non-overlapping time periods of different lengths and months, there was still general agreement with the freeboard distribution. The mean (standard deviation) difference between CryoSat-2 and ICESat freeboard is 2.94 cm (9.23 cm). The fact that the largest differences between CryoSat-2 and ICESat occur in regions of known thick snow depths could signal a difficulty of the algorithm over the thickest snow, suggesting an area for future improvements to the model. In general, this retrieval algorithm shows promise that snow freeboard can be measured from CryoSat-2 alone.

Though the retrieved air-snow interface elevation and snow freeboard closely resemble that from independent measurements, the retrieved snow-ice interface elevation appears to be larger than expected. Calculated snow depth, therefore, is lower than typically expected throughout most of the Antarctic sea ice cover as compared to in situ and passive microwave data. Due to strong attenuation of radar returns from brine layers within the snow pack (Nandan et al., 2017), it may not be possible to retrieve the actual snow-ice interface from a Ku-band altimeter in some regions of the Antarctic. However, the region near the Antarctic coast in the Western

Pacific Ocean (Fig. 11) displays snow depths that are much closer to expected, signaling the possibility of snow depth retrieval under certain ice types and conditions. More work is needed to understand why this region shows near-realistic snow depths while other regions with similar ice characteristics (e.g. positive ice freeboard in the western Weddell Sea) do not.

Overall, this study has expanded the functionality of CryoSat-2 as a tool for observing the snow freeboard of Antarctic sea ice, adding to the existing studies retrieving radar freeboard (Schwegmann et al., 2016) and ice freeboard (Paul et al., 2018). In September 2018, CryoSat-2 was joined in space by ICESat-2, NASA's second-generation satellite laser altimeter system (Markus et al., 2017). These coincident altimeters will provide the ability to observe the polar regions like never before. For this work specifically, ICESat-2 data will be used as both a comparative measure – for direct monthly comparisons of snow freeboard – as well as an initial guess for the waveform fitting model. These new measurements of air-snow interface elevation and snow freeboard from ICESat-2 will help to further validate this retrieval algorithm.

Future work will look into combing these CryoSat-2 snow freeboard measurements with those from laser altimetry to produce an ICESat-CryoSat-2-ICESat-2 time series of snow freeboard in the Antarctic. This reconciled laser-radar altimetric record of snow freeboard would span 15+ years from 2003 throughout the lifetime of ICESat-2, providing a long and robust dataset that could be used in other studies of sea ice. Together with ESA's Climate Change Initiative dataset combing CryoSat-2 and Envisat (Schwegmann et al., 2016; Paul et al., 2018), these long term datasets could lead to improved retrievals of sea ice thickness and an enhanced understanding of sea ice in the Antarctic.

**9 Author contribution**

N.K. developed the framework model and fitting code. S.F. adapted the code and carried out the analysis. S.F. prepared the manuscript with contributions from N.K.

**10 Competing interests**

The authors declare that they have no conflict of interest.

**11 Acknowledgements**

The authors would like to thank the European Space Agency for providing data from CryoSat-2, as well as the reviewers for their constructive feedback. This work is funded by NASA's Airborne Science and Cryospheric Sciences Programs.

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

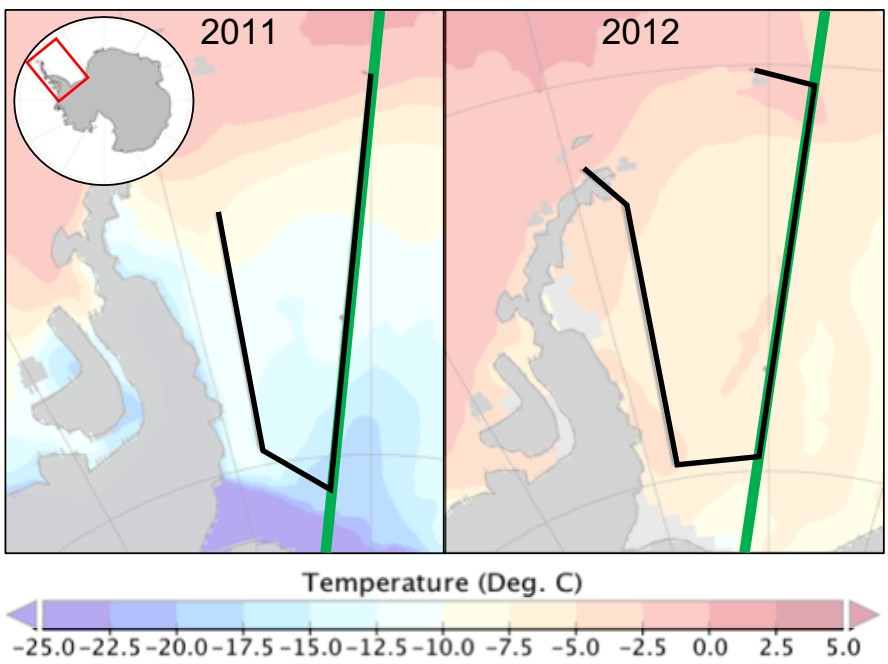

**Figure 1:** Maps of the Operation IceBridge 13 October 2011 (left) and 07 November 2012 (right) Sea Ice Endurance campaign flight paths (in black) along with the contemporaneous CryoSat-2 ground track (in green). Flight paths are overlaid on hourly average sea ice surface temperatures from MERRA-2 at the midpoint time of the IceBridge flight (MERRA-2 data from: doi: 10.5067/Y67YQ1L3ZZ4R).

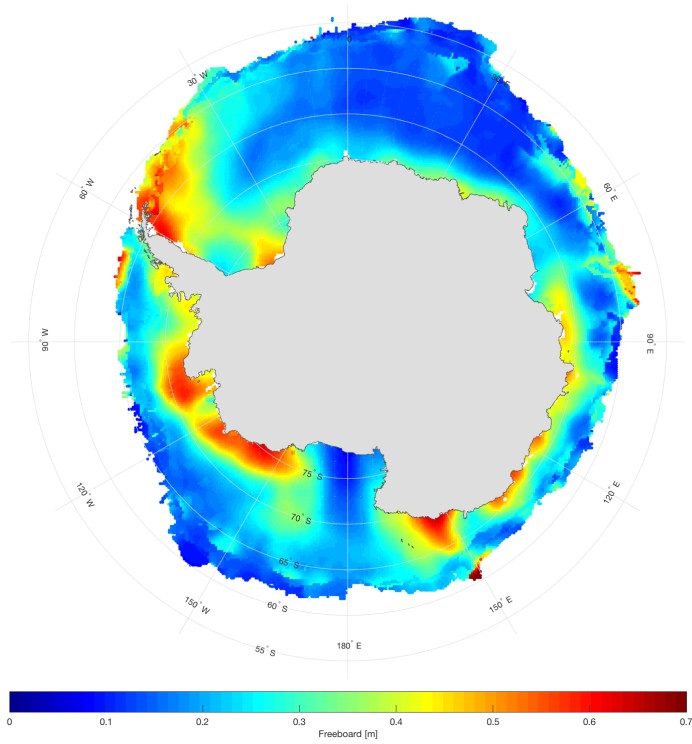

**Figure 2:** ICESat austral spring mean freeboard, consisting of measurements taken in October and November 2003-2007.

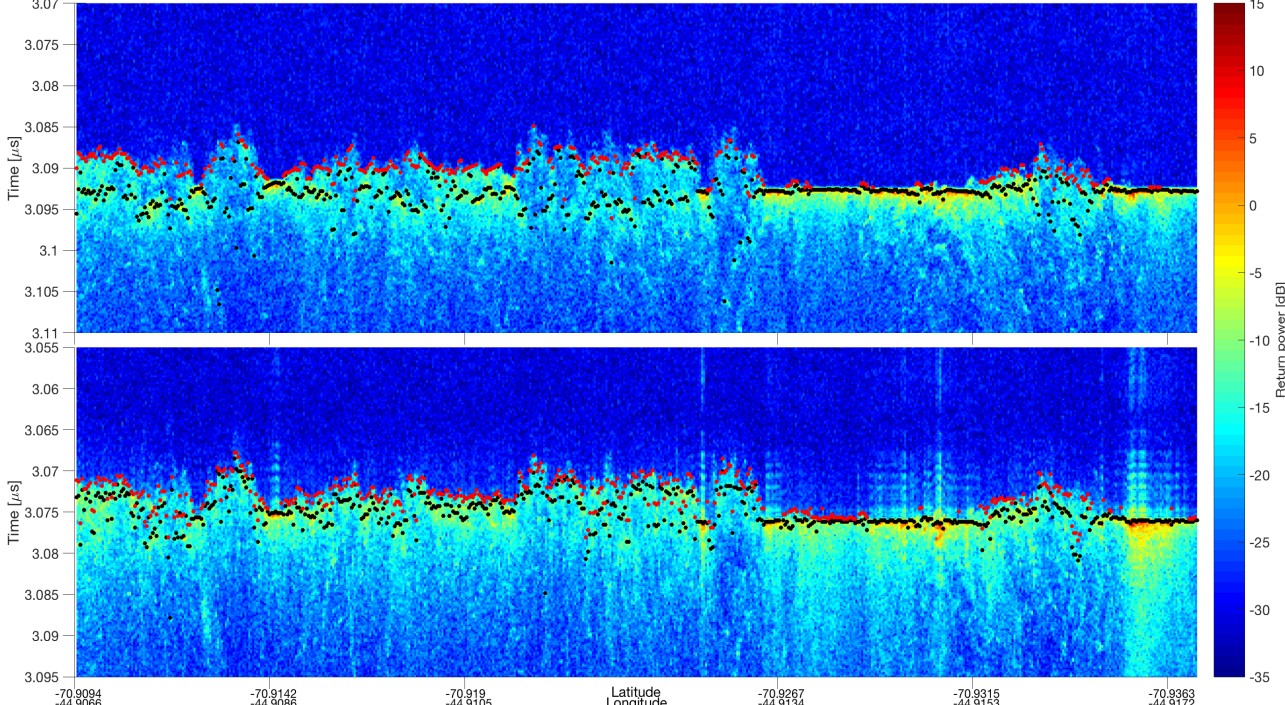

**Figure 3:** Example echograms from Operation IceBridge snow radar (top) and Ku-band radar (bottom) taken from the November 2012 Sea Ice Endurance campaign. Black points denote locations of maximum power and red points denote the first location where the power rises 10dB above the noise level, both found from the peak-picking algorithm discussed in text. The length of the transect covered in this echogram is 3.02km. The mean (standard deviation) noise level for the snow radar is found to be -29.1 dB (1.39 dB) while the signal level at the air-snow interface is found to be -16.8 dB (1.47 dB). For the Ku-band altimeter, the noise level is found to be -30.3 dB (1.32 dB) while the air-snow interface signal level is found to be -17.9 db (2.09 dB), showing the surface return is well above the noise for both instruments.

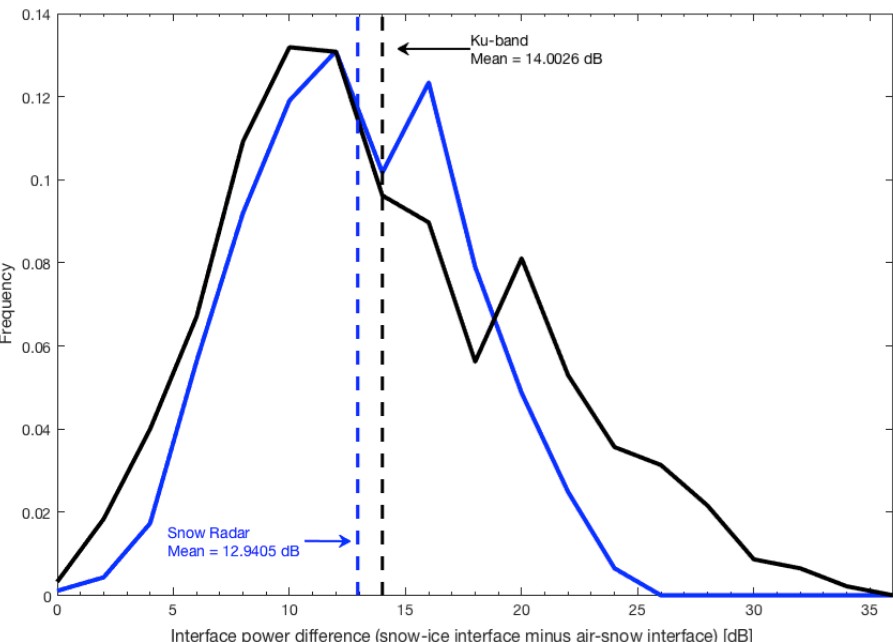

**Figure 4:** Frequency distributions of the difference in air-snow interface power from snow-ice interface power taken from the November 2012 IceBridge Sea Ice Endurance campaign. The blue curve represents the snow radar, while the black curve represents the Ku-band radar. Note that the locations of the air-snow and snow-ice interfaces are approximations found from the peak-picking algorithm (Fig. 3) and are not exactly the expected backscatter coefficients from the two layers.

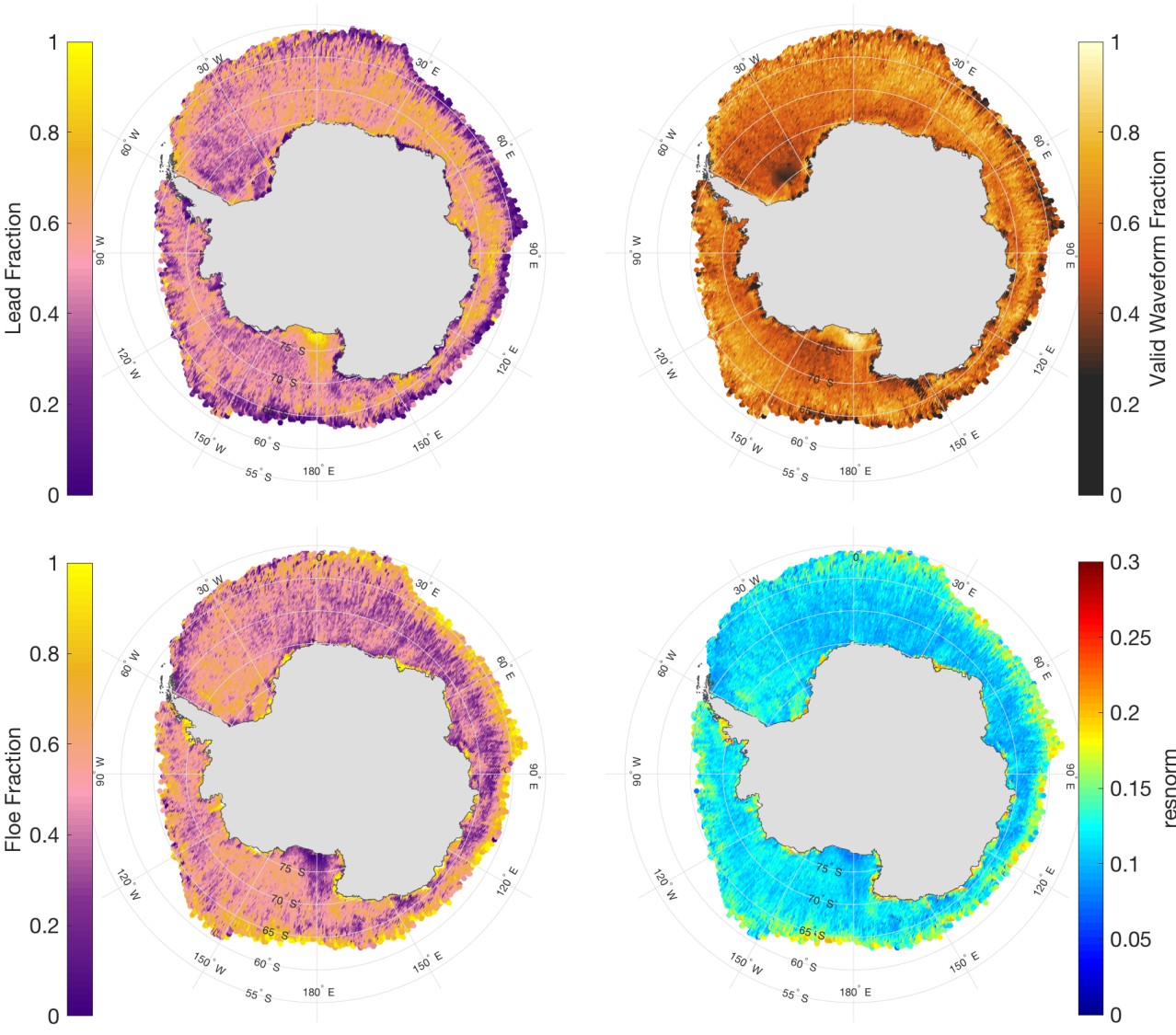

**Figure 5:** October 2011-2017 average maps of lead-type waveform fraction (top-left), floe-type waveform fraction (bottom-left), valid waveform fraction (top-right), and resnorm value (bottom-right).

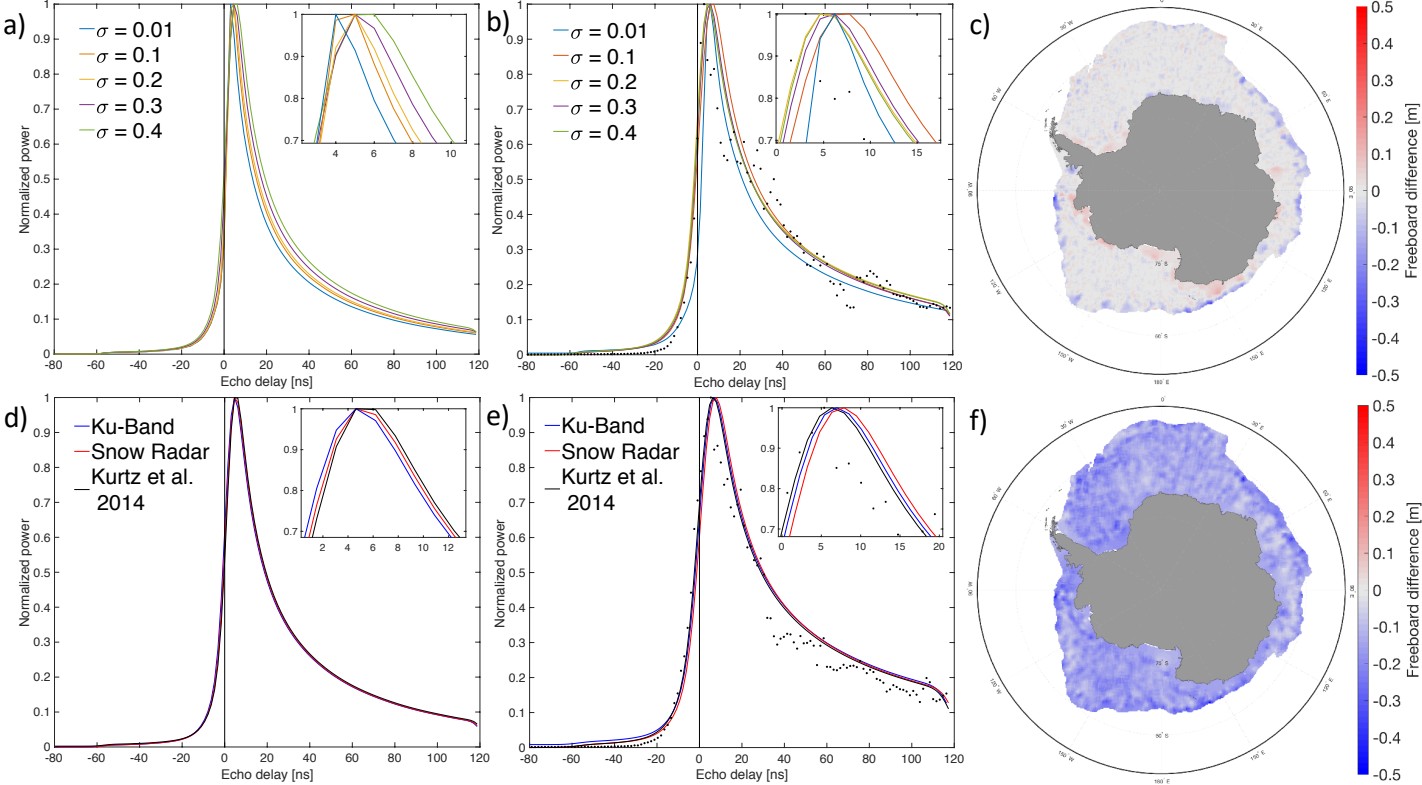

**Figure 6:** A sensitivity study of two initial guess parameters: the standard deviation of surface height, σ, and the total backscatter, $σ^0$. (a) modeled waveform (before fitting) varying the initial guess value of σ between 0.01 (very smooth surface) and 0.4 (rough surface). (b) waveforms fit to CryoSat-2 data varying the initial guess value of σ between 0.01 and 0.4. (c) October 2016 average freeboard difference: σ = 0.35 as the initial guess – σ = 0.15 as the initial guess. (d) as in (a) using three different backscatter parameterizations taken from the OIB Ku-Band radar profile, Snow Radar profile, and Kurtz et al. 2014. (e) as in (b) with the three different backscatter parameterizations. (f) as in (c) showing Kurtz et al. 2014 backscatter as the initial guess – Ku-Band backscatter as the initial guess. Inlaid plots are zoomed in on the waveform peaks. The methodology for freeboard calculations is explained in later sections of the paper.

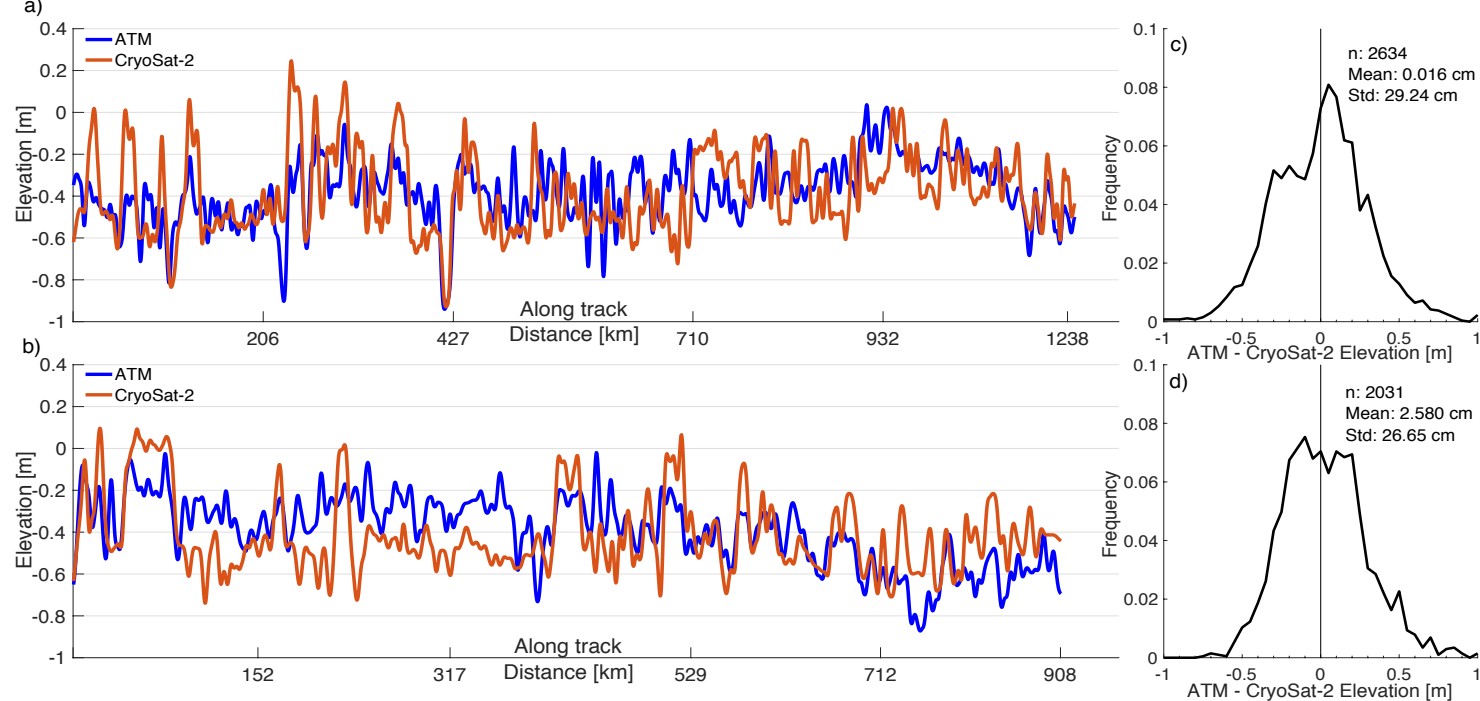

**Figure 7:** Surface (air-snow interface) elevation profiles of Operation IceBridge ATM (blue) and CryoSat-2 (orange) from the October 2011 (a) and November 2012 (b) campaigns. Frequency distributions of the elevation difference (ATM – CryoSat-2) along the 2011 (c) and 2012 (d) profiles are also shown. The mode of the differences is 0.025m in 2011 and -0.24m in 2012. The 2011 profile contains measurements from -63.99° N, -45.11° W to -75.04° N, -49.33° W while the 2012 profile contains measurements from -66.14° N, -43.31° W to -74.25° N, -46.46° W.

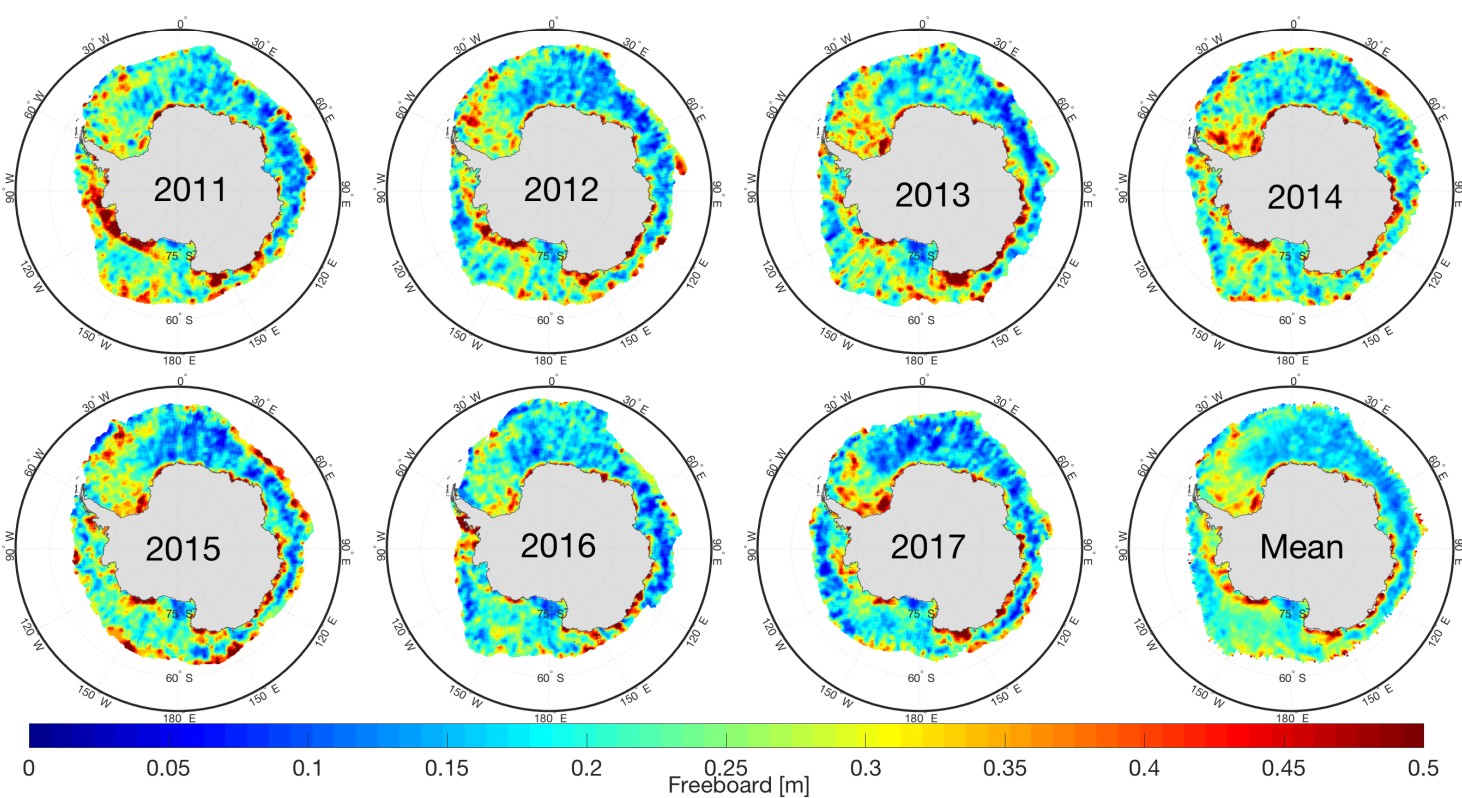

**Figure 8:** October monthly average snow freeboard from 2011-2017, as well as the mean of all years, found using this retrieval method.

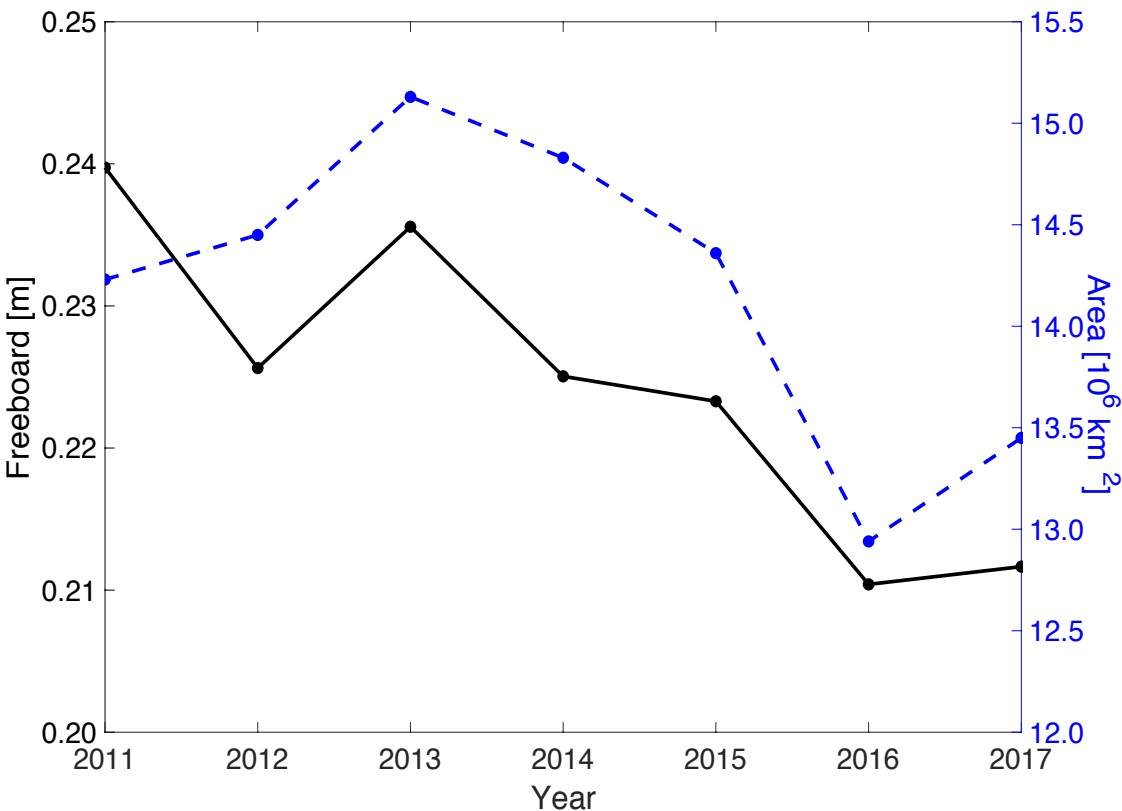

**Figure 9:** October monthly average Antarctic snow freeboard (black) and total sea ice area (blue) for reference. Sea ice area data are gathered from NSIDC (Fetterer et al., 2017) and can be found at nsidc.org/data/G02135.

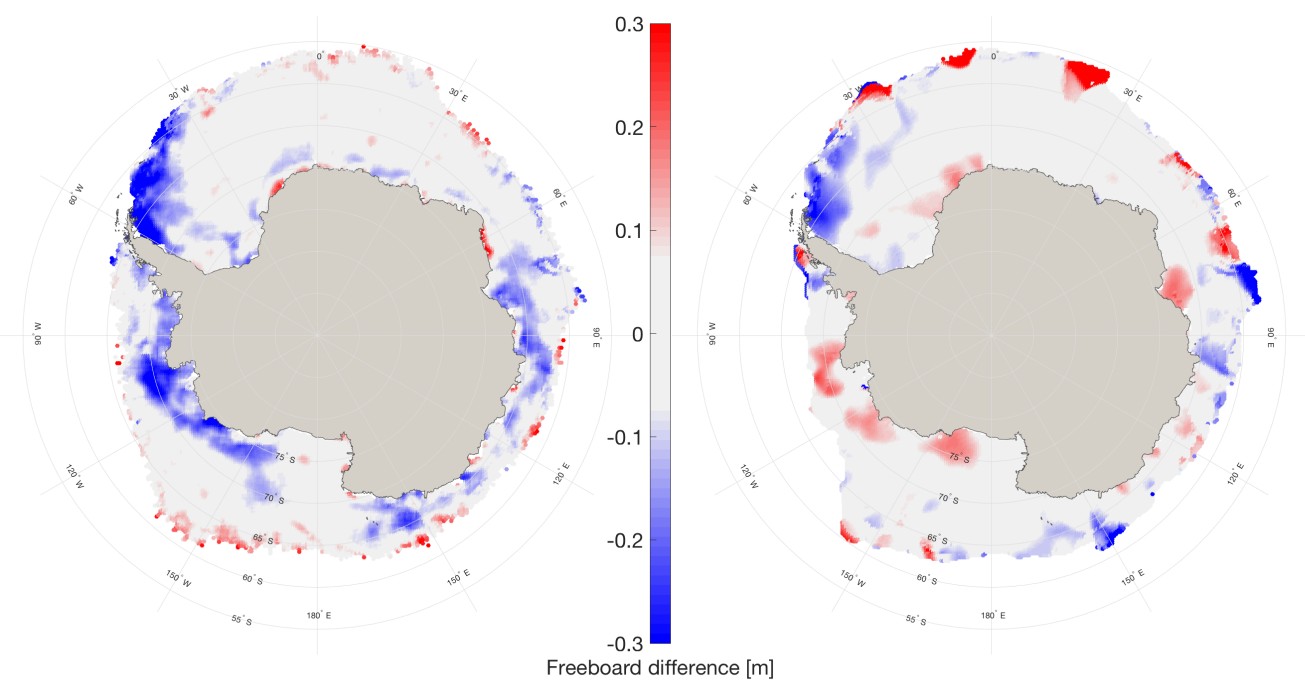

Freeboard difference [m]

**Figure 10:** Snow freeboard differences showing (left) CryoSat-2 October 2011-2017 average minus ICESat spring 2003-2007 average and (right) ICESat spring 2006 average minus ICESat spring 2003-2007 average. 2006 is included as an example year to highlight the interannual variability in the freeboard distribution.

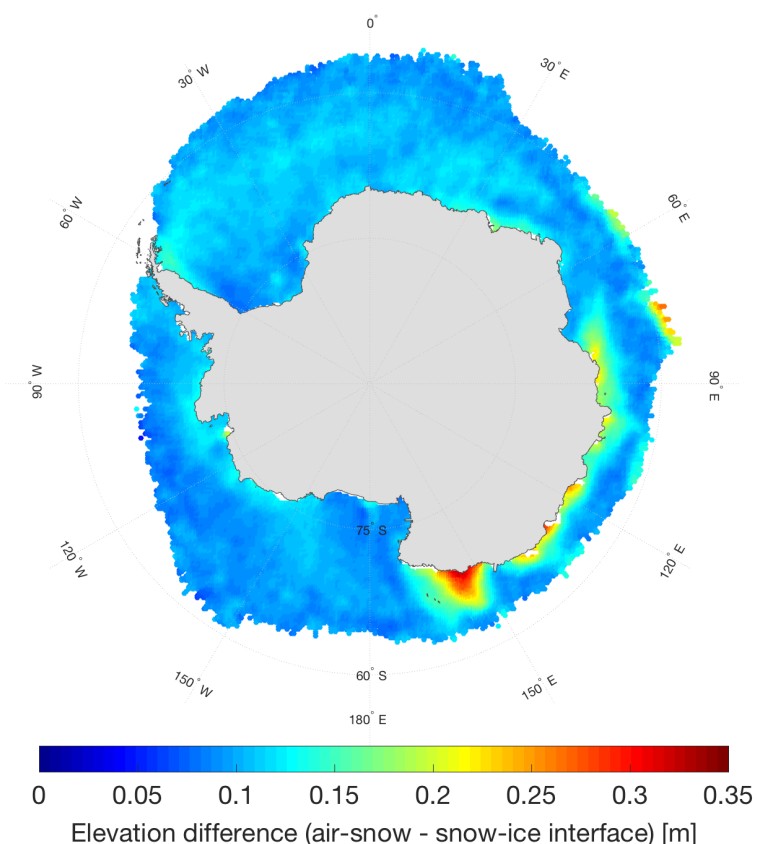

**Figure 11:** October 2011-2017 average difference between the retrieved air-snow and snow-ice interfaces as an exploration into the potential retrieval of snow depth.