# Peer review of "Retrieval of snow freeboard of Antarctic sea ice using waveform fitting of CryoSat-2 returns"

_The Cryosphere, 2018_

## Referee Comment (RC1) · Anonymous Referee #1 · 24 Oct 2018

Review of

Retrieval of snow freeboard of Antarctic sea ice using waveform fitting of CryoSat-2 returns

by

Fons, S. W., and N. T. Kurtz

Summary: This paper investigates whether the considerable scattering of satellite radar altimetry at Ku-Band, namely CryoSat-2, that can be expected to occur at the air-snow interface can be exploited to estimate the elevation of the air-snow interface

relative to the ocean surface and hence get an estimate of the total (sea ice + snow) freeboard. To do so a two-layer physical model is used together with least square fitting to obtain a fitted waveform to CryoSat-2 Level 1B data from which the elevations are obtained for lat winter / spring October months of 2011-2017. CryoSat-2 elevations are compared with observations from Operation Ice Bridge airborne topographic mapper for two quasi-coincident OIB-CS-2 overflights, one in 2011, one in 2012. Total freeboard is computed and averaged over the entire period 2011-2017 for the entire Antarctic and discussed and compared with ICESat total freeboard maps. Also, the potential to combine air-snow and ice-snow interface elevations for snow depth on sea ice retrieval is tested. While first evaluation results are promising and suggest that total freeboard derived from CS-2 could potentially be used complementary to ICESat and ICESat-2 data more work is required to better understand the observed differences between CS-2 total freeboard and independent data.

I find this an interesting and important contribution to the existing literature and suggest publication of the research results - provided that the authors take into account the various, partly substantial, suggestions for a major revision of their manuscript. I list my major concerns below in the general comments.

General comments: GC1: The introduction lacks to present the state-of-the-art of sea-ice and/or total freeboard retrieval in the Antarctic. Several studies exist that are using radar or laser altimeter data for this kind of retrieval. In addition, the introduction lacks to present the state-of-the-art of freeboard-to-thickness conversion and inherent problems and uncertainties. While mitigation of the former lack is required to understand why it might make sense to try to derive total freeboard also from CS-2 data, mitigation of the latter lack is required to understand your attempt to retrieve snow depth on sea ice as well.

GC2: I don't find the interpretation of Figures 3 and 4 particularly convincing as a motivation why there is substantial(ly more) information about the air-snow interface in the echograms of the snow radar and the Ku-Band altimeter. See my respective

specific comments.

GC3: Even though Figure 6 and the interpretation is intended to stay qualitative (my guess), I strongly suggest to discuss these results in more detail. Putting more emphasis on increasing the credibility of the elevation estimates at this stage is very important in my eyes. You have the unique opportunity to have quasi-coincident air-borne and space-borne measurements. That's luxury and I have to admit that I am a bit disappointed that you do not exploit this luxury situation further. If I'd be allowed to recommend something, then I would i) quantify the temporal and spatial differences in the tracks and try to investigate whether a correction towards a better spatiotemporal match is worth an effort, ii) collocate the tracks with ice-type information (it might be sufficient to figure out where first-year ice and perennial ice was present), iii) collocate the tracks with meteorological information, e.g. from ERA-Interim or MERRA-2 or perhaps even from one of the higher-resolving weather forecast models to figure out whether air temperatures have been close to 0degC and/or whether and what kind of precipitation potentially occurred (Ideally you have a look at the meteorological conditions of not just the day of the coincident measurements but also of a 1-2 weeks period before to catch potential melt events and hence snow metamorphism near the air-snow interface.). See also my respective comments for Figure 6 and its interpretation.

GC4: The interpretation of Figures 7 through 9 would also very much benefit from a more critical discussion which should also involve more work done by other researchers. I find a lack of attempts to explain the differences observed, e.g. in Figure 9. See my specific comments to these figures.

Specific comments:

Abstract: I suggest to add the standard deviation or Root-Mean-Squared difference in addition to the mean difference values given. If computed, also modal values of the difference would allow to give the obtained values more credibility.

Page 1 L30: I guess, since you are focussing on Antarctic sea ice it would not hurt

do use citations referring also to the albedo observed over Antarctic sea ice: Brandt RE, Warren SG, Worby AP and Grenfell TC (2005) Surface albedo of the Antarctic sea ice zone. J. Climate, 18(17), 3606–3622 (doi: 10.1175/JCLI3489.1) and Zatko and Warren, Annals of Glaciology 56(69) 2015 doi: 10.3189/2015AoG69A574

Page 2 Line 2: I suggest to cite Comiso et al., J. Climate, DOI: 10.1175/JCLI-D-16-0408.1 instead of Beitler 2014; the former is a peer-reviewed paper.

Line 6: Please add "Antarctic" or "Southern Ocean" to make clear that these ship-based observation based sea-ice thickness data set is valid there but not general in the polar regions.

Line 5-13: - Is there a reason why you refer to multiyear ice only for the Arctic? - Is there a reason why you refer to sea-ice thickness in the Arctic only while for the Antarctic you refer to sea-ice thickness in volume? Is the sea-ice thickness retrieval in the Antarctic more accurate so that it makes sense to also derive the volume?

Line 14-20: - I am not sure I like the mentioning of Kwok et al. (2009) and Kurtz and Markus (2012) as the role models for sea-ice thickness measurements from active satellite sensors. Since you are basically referring to the principle, wouldn't it be sufficient to simply write what you wrote without these two references? I I guess my dislike comes from the fact that there have been earlier papers that describe how laser altimetry (which is the main focus in this paragraph) can be used to get an estimate of the total freeboard of snow-covered sea ice: Kwok et al. (2004) or Kwok et al. (2006) for the Arctic and Zwally et al. (2008) for the Antarctic.

L21-30: - I suggest to re-organize the sentences starting in Line 24 to avoid that sea-ice freeboard is used before being explained. How about you write along these lines: "... 2010-2012. The difference between laser ... [continue until Line 29] ... above the sea surface, known as the "sea-ice freeboard", and is used to calculate sea-ice thickness applying appropriate assumptions (see previous paragraph)."

[Figure]

L31-39: - I suggest to expand rightaway in Line 31: "by the depth and variable vertical structure of the snow on top ..." - In Line 32, I suggest to add that it is not simply more precipitation but "... more and more frequent precipitation ..." - Line 34: "sea ice down near the" –> perhaps better: "sea-ice surface down near or even below the" - I suggest to break in Line 38 for a new paragraph, starting with "While ... ". - Is there perhaps also the chance that you quantify how large or weak the scattering at the air-snow interface is compared to that at the ice-snow interface? This could make your motivation stronger about why it might be reasonable to look for the snow surface scattering contribution even in Ku-Band. I strongly suggest to seek for evidence in the literature about the possible strength of the snow surface backscatter at Ku-Band (at nadir) to underline that it is physically reasonable to use CS-2 SIRAL returns for snow freeboard retrieval. I am stressing this because there exists literature in which one aims for snow-depth on sea ice retrieval by confidently assuming that Ku-Band penetrates to the ice-snow interface and combining it with a Ka-Band radar such as from SARAL AltiKa (Guerreiro et al., 2016). Your attempt is clearly conter-acting their assumptions.

Page 3: Line 10: "builds off" ?

Line 22: I don't understand the mentioning of the "originally 128". What is this for?

Line 34-37: The motivation for choosing data from October is clear. You could have stated why you did not also use data from November. Most of the ICESat spring measurement periods last well into November. Here you state years 2003 to 2009 for ICEsat as years with measurements but actually using you are only data from 2003 to 2007. I can understand that the main motivation for this is to use the data produced by one of you. But from NSIDC and potentially also from University of Hamburg you could possibly have obtained ICESat freeboard data for 2003 through 2009, i.e. from an equally long period as you have CS-2 data from. You stated yourself explicitly, that "Seven years of data allows for a longer-term average to be computed"

Page 4: Line 1: - It would be helpful for a better understanding of Figure 3 to mention

the frequency range of the FMCW snow radar. - "First, ..." –> Where is the "Second"?

Line 8-12: Is this a gridded product? If yes, which grid resolution does it have and what is done to fill the gaps between the ICEsat overpasses?

Line 13-16: You are using sea-ice concentration data obtained with the NASA-Team algorithm. While when choosing a 50% sea-ice concentration threshold is might not really matter which product to use there is published evidence that the NASA-Team algorithm often severely under-estimates sea-ice concentrations in the Antarctic compared to the truth - particularly in late winter / spring. You could avoid students and early-career scientists being trapped by your choice by choosing a more appropriate sea-ice concentration product right from the beginning, i.e. based on the Comiso-Bootstrap algorithm or the Eumetsat OSI-450 algorithm.

Line 18: "... altimetry tend ..." –> I suggest to insert: "for ice freeboard retrieval"

Line 20: You could cite Willat et al. (2011) here.

Line 35/36: "This result is expected, as it means that the scattering power from the air-snow interface is closer in magnitude to that of the snow-ice interface in snow radar returns" I do understand your conclusion from the smaller difference in power (about 13 dB for snow radar and 14 dB for Ku-Band altimeter) but I don't understand why this is expected. Lets assume for simplicity that the peak power is 20dB at the ice-snow interface for both instruments. Then the power at the air-snow interface would be about 7 dB for the snow radar and 6 dB for the Ku-Band altimeter and with that the power at the air-snow interface would be SMALLER at that frequency from which you assume that the backscattering at the air-snow interface is LARGER. How does this fit together?

Page 5: Line 2: At the end of this paragraph interpreting Figures 3 and 4 I have a few questions. i) How accurate are the two instruments with respect to the dB values shown? ii) How relevant is the similarity in the histograms shown in Figure 4 with

respect to the shared mode at about 11dB while at the same time the histogram shows secondary modes at 16dB (snow radar) and 20 dB (Ku-Band). In addition I have a few comments: iii) What were the meteorological conditions during that flight? Can we expect homogeneous snow properties in terms of snow wetness etc. iv) How would Figures 3 and 4 look for the October 2011 campaign? Would they result in the same result? v) What is the length (in kilometers) of the transect (or echogram) shown in Figure 3? vi) The Ku-Band altimeter histogram in Figure 4 has a substantially longer right tail with high dB values. It is almost certain that these values are responsible for the 1dB difference observed between the snow radar and the Ku-Band altimeter data. Are these particularly large differences the result of a particularly low power at the air-snow interface compared to the peak power or is this the result of peak powers being generally elevated at Ku-Band compared to the snow radar? vii) What explains the larger time difference between the locations denoted by red and black points in Figure 3 for snow radar echogram compared to Ku-Band? viii) What is the source for the staggered echos above the air-snow interface at Ku-Band? Such echos are not at all present for the snow radar data.

Line 21: What is "n"?

Page 6: Line 2: I guess "thicker snow depth" and "scattering effects from the snow surface" are not as much linked with each other as scattering effects from the snow volume. What makes a thicker snow cover to have more surface scattering than a thinner snow cover?

Line 27: Why "Though"?

Page 7: Line 16: What is the motivation to only use a different initial guess for alpha in case resnorm is too high after the first fitting attempt?

Page 8: Line 8: Are these PP and SSD thresholds also taken from Laxon et al. (2013)?

Line 12: "seasonal average freeboard datasets" –> perhaps better "datasets of the

seasonal average total freeboard"?

Line 15: A very good place to cite the paper by Kwok and Maksym, 2014 in J. Geophys. Res., doi:10.1002/2014JC009943

Lines 23-25: Please explain how the OIB data area used. Did you take data from both flights? Did you average over all valid points? What is meant by "respective surfaces"?

Two final question at the end of this section: What happens in the special case of a snow free ice floe? What happens in case of a wet snow surface, where penetration of the Ku-Band into the snow cover is almost zero?

Line 32: "found" –> perhaps better "computed" or "derived"?

Page 9: Lines 7-13 & Figure 6: - Please provide a measure of the total distance along the tracks shown in Figure 6 a) and b). This would make referencing to certain feature more easy in addition to simply providing an easier interpretation of the spatial scale we are looking at. - I suggest to add a vertical line at zero difference ATM minus CS-2 in images c) and d). - What is the average difference in successive measurements in images a) and b)? - Suggest to use the same y-axis scaling for a) and b) for a better visual comparability of the elevation variations. - I'd say that the overall agreement, i.e. the large-scale agreement is better for 2011 than 2012. - For 2011 the mean is very close to zero, right. But the modal value is between 5 and 10 cm with CS-2 underestimating ATM elevation. - CS-2 elevations quite often exhibit strong variations in magnitude; in 2011 more during the first third of the track, in 2012 during the first two thirds of the track. These strong variations (or jumps) are as large as about 40 cm and except in one case do not have a counterpart in the ATM elevations. Please try to give an explanation to these. - While in 2011 the large-scale agreement is quite good (you could even stress this impression by adding elevation profiles with large-scale smoothing applied), in 2012 there appears to be a systematic under-estimation of the ATM elevation by CS-2. Please try to give an explanation to these as well. - You argue that differences between the two elevation data sets might be caused by "initial

temporal and spatial discrepancies between the two data sets. Would you be able to quantify these differences? Would it make sense to do a correction of the track of one sensor with respect to the track of the other sensor? - You write that both datasets "appear to detect similar locations of troughs and ridges along the flight line". I don't find this statement particularly convincing because there are also many cases where troughs in one dataset and ridges in the other dataset coincide.

Lines 19-33: - Line 25: "fewer than five data points" –> What is the distance of successive data points? How many data points would fall into one 25 km grid box at maximum, i.e. diagonal crossing? Can you comment on the data density as well? How many CS-2 overpasses or days with CS-2 overpasses in one grid cell do you have in one month? - Lines 29/30: "Any points within each grid box ..." –> Could it be that this sentence should be placed before the previous sentence? I am asking because the previous sentence already describes the method used at grid level. - Line 32: "are smoothed" –> Why is this? Why do you do that? Is it because of the gaps between the overpasses? Please state so in the paper.

Page 10 Paragraph ending in Line 7: - While there is not too much work yet about freeboard distribution from radar altimetry in the Antarctic I still suggest that you consider comparing your results with the results published by Giles et al. (Geophys. Res. Lett., 2008), Schwegmann et al. (Annals of Glaciology, 2015), and Paul et al. (TC, 2018). - While the work of Nghiem et al. (2016) is really interesting and certainly not invalid over parts of the Antarctic MIZ I suggest that you also take into account (and at least mention if not discuss) the potential effect of ocean swell, lower CS-2 data density and hence a larger representativity error, and ice types being different in the MIZ than in the pack ice; a large fraction of the Antarctic MIZ is formed by the often several hundreds of kilometers of pancake ice or cake ice or first-year ice with small floe sizes (< 100 m) for which I doubt that CS-2 is going to provide reasonable elevation and hence freeboard estimates. - I note that the distribution of total freeboard shown in Figure 7 is quite patchy and contains several artificial south-north oriented freeboard variations (possi-

bly caused by sampling issues). I note that the freeboard in the southern Ross Sea is indeed lower than further north. However, given the fact that this is an area of extensive new-ice formation and export paired with low precipitation and hence thin snow cover, the freeboard values shown are certainly at the higher end of what is typical there - if not a proper overestimation. Sea-ice thicknesses in the southern Ross Sea are 20 to 50 cm ... total freeboards (without snow) therefore in the range of between 2 and 5 cm and not between 10 and 20 cm as indicated in the maps.

Lines 8-15 & Figure 8 - "smallest measured freeboard" –> perhaps better "smallest measured mean October freeboard" - 25.77, 27.6, 12.97 ... I suggest to give these figures with the same number of digits, i.e. 25.8, 27.6 and 13.0. - Showing the mean total freeboard together with the sea-ice area is certainly fine even though, as you stated correctly, it is not too clear why you find a good correlation between these two quantities. However, instead of the sea-ice area one could plot other variables as well. One would be the standard deviation of the mean total freeboard as a measure of the scatter of the mean values. A second one would be to show either the number of 25 km grid resolution grid cells with valid CS-2 data or even the number of individual valid freeboard (or elevation) measurements. Since you have many gaps in the original CS-2 data it would potentially be a very interesting additional information. - I note that the maximum inter-annual difference of the mean October freeboard is 2 cm. Is this within or outside the retrieval uncertainty?

Lines 17-25 and Figure 9: - I suggest to color the open water in a grey tone (different than Antarctica of course) to ease discrimination between areas with differences close to zero and open water. - I suggest to reduce the range of the differences shown to +/- 30 cm to show more details. The way the range is chosen currently only reflects the larger differences. - I find it essential that you mention three things in your discussion of this Figure: i) the larger number of years for CS-2 (7 instead of 5), ii) the fact that the ICESat data cover different time periods with at least 2 of the five years have a substantial if not dominating overlap in time with November and hence conditions changed

towards spring (As far as I recall you analysis you did make the effort the average CS-2 from exact those dates from which also ICESat measurements exist.), iii) the fact that we look at years 2011-2017 for CS-2 but 2003-2007 for ICESat, i.e. two different, not overlapping time periods. While this might not have an effect it needs to be stressed once more in the context of this discussion. Finally, iv) one could ask whether you used the same method for averaging the CS-2 data (and filling gaps, extra- or interpolating gaps) than was done in Kurtz and Markus, 2012? Because of these four issues I warmly recommend to delete the last sentence in Lines 24/25. - When talking about a difference of only 1.9 cm: What are the uncertainties in monthly mean freeboard from CS-2 and from ICESat? Is the difference about the uncertainty? - You do not make any attempt to explain the highlighted negative freeboard differences CS-2 minus ICESat in the Weddell and Amundsen Seas. Why? - How do your results compare to the work of independent researchers: Yi et al., 2011, Kern and Spreen, 2015, Kern et al., 2016, Li et al., 2018?

Lines 27-36, Figure 10: - Figure 10 is indeed quite interesting because the highest "snow depths" are not observed in the Weddell Sea but on East Antarctic sea ice. Puzzling. This is even contradicting your own work (Kurtz and Markus, 2012), where the freeboard maps shown are assumed to represent the snow depth while assuming sea-ice freeboard to be zero. In that work maximum freeboard and hence "snow depth" was observed in the Weddell Sea. What is further interesting is that the "snow depth" is nowhere considerabler smaller than 10 cm - even not in the southern Ross Sea where there is little or no snow on the young sea ice. - I would have found it again very useful, if you would have related the results shown in Figure 10 also with other work, i.e. snow depth based on satellite microwave radiometry (Markus et al., various) or based on ICESat data (Kern and Ozsoy-Cicek, 2016).

Page 11: Lines 1-4: That the peak-picking algorithm provides snow depth along that flight line which is within 10% of the values published by Kwok and Maksym (2014) is an encouraging result and should be highlighted more. Lines 4-6: I am pretty sure

that this is a frequency issue and not an issue of bandwidth or footprint size: The snow radar used on OIB is a 2-8 GHz radar, right?, while CS-2 operates in Ku-Band. With the "correct" snow conditions it is very likely that CS-2 does not penetrate down to the ice-snow interface, explaining the considerably lower "snow depth" value estimate from the two elevations. Actually, the OIB snow depths are potentially even higher because of the difficulties to retrieve snow depth in areas of deformed sea ice and on multiyear-like ice reported elsewhere. Comment: Again I doubt that the precision and accuracy of the data warrants to give mean values with 3 digits = millimeter precision here. I guess 0.29 m, 0.26 m and 0.15 m would do it.

Line 8: I suggest to delete "slightly". It is considerably greater. Line 12: "validating" –> perhaps better "evaluating" or even only "understanding". Line 13: "to better understand the snow depth distribution on sea ice" –> perhaps better: "to use it together with the air-snow interface for snow depth on sea ice estimation."

Lines 15-27: - Line 16: "air-snow interface of sea ice" –> perhaps better "air-snow interface of snow on sea ice". - Line 20: "validate" –> "evaluate" - Line 24: "data from comes from" ...? - Line 22-27: I strongly suggest you revise these conclusions based on the additional analysis, interpretation and discussion that is recommended in the general comments. In particular, figures for standard deviation and potentially also uncertainties should be given in addition to the mean values. One can have a mean value close to zero with one part of the data pairs having -50 cm difference and the other part having +50 cm difference ...

Lines 28-33: - Line 28: "retrieved ice freeboard" –> you did not really retrieve ice freeboard, did you? You computed the snow depth from the difference between air-snow interface elevation and snow-ice interface elevation. - Line 29: "lower than typically expected" –> since you did not show any other results about snow depth on Antarctic sea ice - expect for the case of East Antarctic sea ice - it is difficult to follow this statement. - Line 31: I agree about the potentially wide-spread flooding of Antarctic sea ice but sea ice is mostly flooded when the ice-snow interface is submerged below the sea level and in that case an ice-snow interface does not exist in that sense anymore. If it still exists, e.g. through to lateral flooding it is possibly close to zero. It might therefore be more correct to again refer to sea water and brine wicked up into the snow, creating a saline snow - non-saline snow interface which is possibly the interface seen by Ku-Band. Question: For the location of the air-snow interface you have a-priori information from seasonal mean ICESat snow freeboard. For the ice-snow interface you don't have any a priori information, do you? This could be one explanation for the sub-optimal performance with regard to detecting the ice-snow interface as well. - Line 34: I guess it would be fair to cite the already existing literature about using CS-2 data for Antarctic freeboard (sea-ice thickness) retrieval: Paul et al., 2018, and change "... observing Antarctic sea ice." into something like: "... observing Antarctic sea ice with satellite radar altimetry in addition to Paul et al. (2018)." Maybe there is even more work out using CS-2 data in the Antarctic? Please check!

Page 12, Line 7: Again I think it would not be too bad to add the work of other authors here to avoid the impression that you are the first on this field: "... for improved retrievals of Antarctic sea ice thickness, complementary to sea-ice thickness retrievals based on the 15+ years long time series of combined Envisat - CryoSat-2 freeboard estimates (Paul et al., 2018)."

Page 23: Line 23: Giles et al. This paper was in Geophys. Res. Lett. and not The Cryosphere

---

## Referee Comment (RC2) · S. Hendricks (Referee) · 2 Nov 2018

In their paper "Retrieval of snow freeboard of Antarctic sea ice using waveform fitting of CryoSat-2 returns", the authors develop and apply an algorithm to obtain snow freeboard of Antarctic sea ice using waveform fitting of CryoSat-2 data. The waveform fitting is based on a forward model from earlier work of one of the authors and the main work here has been the to include backscatter from multiple interfaces (air/snow and snow/ice) in combination with snow volume backscatter for the application of sea ice in the southern hemisphere with its higher and more complex snow load. The authors compare the results with airborne validation data and earlier ICESat laser altimeter results and conclude that their algorithm can be used to obtain snow freeboard with the CryoSat-2 during the maximum austral sea ice extent in October. They also investigate the potential to retrieve snow depth from CryoSat-2 waveforms, but do not find realistic results except for an area in the East Antarctic sector.

I have been part of the team that produced freeboard maps of Antarctic sea ice from Envisat & CryoSat-2 data in the Climate Change Initiative. We used an empirical re-tracking scheme, which made it difficult to include the contribution of snow backscatter in the freeboard algorithm without prior knowledge of its impact on waveform shape. We are acutely aware of this deficiency in the CCI sea ice thicknesses for the southern hemisphere, and it is commendable that the authors attempt to overcome this issue. Therefore this study is for me a very welcome and novel contribution to the field of re-mote sensing of Antarctic sea ice thickness. The concept is generally sound, but there are a number points that need to be addressed before publication. I have detailed my concerns in the general and specific comments below:

General Comments:

1) It is understandable that maps of snow freeboard is the main objective of this work, but the authors show very little in terms verification of the different algorithm steps. The waveform fitting is based on nine free parameters, but it is not clear to me what the sensitivities are for resolving the snow backscatter properties. E.g. do snow backscatter/depth and surface roughness have an ambiguous impact on waveform shape and thus range? In general, the potential impact of surface roughness changes receives very little attention in the evaluation of the results. The fact that the snow depth resulting from the waveform fitting is unrealistic in wide parts of the study regions shows that there are issues with fully resolving the backscatter processes. A sensitivity study of the waveform forward model would help greatly to assess the skill of the algorithm for different snow conditions. Regional information on the average waveform fit quality would also be of interest to the reader as it might help to identify issues of the algorithm. I find this especially important as the direct validation with the Operation IceBridge ATM

and snow radar data is very limited compared to the spatial and temporal extent of the CryoSat-2 data and the otherwise indirect comparison to ICESat.

2) The conversion of surface elevations into snow freeboard is not state-of-the-art, in parts problematic and the authors risk to undermine the value of their retracker development work. For example, the authors use a surface type classification scheme that was originally designed for Arctic sea ice and an earlier version (baseline-b) of the CryoSat-2 Level-1 data. For the ESA CCI data set we had to define separate waveform parameter threshold for Arctic and Antarctic sea ice. The authors need to show that there is no preferential sampling of surfaces that may introduce a freeboard bias, which can be easily done by providing information on detection rates for lead and floe surfaces and compare those the surface type fractions of the ESA CCI dataset (Paul et al. TC 2018). I would also strongly suggest to compute freeboard per orbit and not by subtracting monthly mean elevations and sea surface height. In our experience the geophysical range corrections in the CryoSat-2 Level-1 product files are not good enough for this approach. Using the instantaneous sea surface height during the orbit will be more reliable and also yield better options for filtering incorrect retrievals.

3) Several earlier studies and datasets that are highly relevant for this topic are not mentioned and the manuscript gives the wrong impression that the work of the authors is the first application of CryoSat-2 for Antarctic sea ice. The authors also state that the impact of snow backscatter on ranging with Ku-Band frequencies is "often ignored" which I certainly do not agree with. It might be a matter of wording, since most operational CryoSat-2 products use the assumption that the freeboard is the average ice freeboard within the footprint. But this is due to the challenges of parametrizing snow backscatter and its temporal and regional variability, not the lack of awareness in the scientific literature. I have suggested references in the specific comments below. The lack of reference to existing publications that specifically deal with CryoSat-2 freeboard retrieval in the southern hemisphere (Schwegmann et al, 2016, Paul et al, 2018) is also unfortunate, as these are a good motivation for this study. There would be added value

if the authors compare their snow freeboards to the freeboard information in the ESA CCI CryoSat-2 data set (see data availability in Paul et al. TC 2018) and demonstrate the improvements and limitations of using waveform fitting.

Specific Comments:

P2L7: I guess you mean "active" in the sense of active microwave sensors respective altimeters in general? I would recommend to use the term "satellite altimeters" instead of "active platforms" throughout the document. (Typo: remove -> remote)

P2L27: Typo: "of off"

P2L28: Beaven et al. 1995 states that the snow/ice interface is the dominated backscatter source. To my knowledge Beaven itself does not imply that cold snow under laboratory conditions is completely transparent for Ku-Band. This is a fine distinction but relevant for this paper.

P2L39: Please rephrase the term "often ignored" as it does not properly reflect the state of the scientific literature. The issue is not lacking awareness of the importance of snow interface or volume backscatter for Ku-Band freeboard retrieval (e.g. Armitage and Ridout GRL 2015, Ricker et al. GRL 2015; Nandan et al. GRL 2017), it is the challenge of getting the temporal and spatial variability from the available waveforms. A better description would be "often not included freeboard retrieval algorithms, especially those depending on an empirical waveform evaluation".

P3L30: The inverse barometric correction is included in the dynamic atmosphere correction. Both corrections should not be applied in combination.

P4L18: See comment above. There are several publications that investigate the impact of snow on CryoSat-2 freeboards and are not cited here.

P7L7ff: At this point it would have been good if the authors had included a sensitivity study using their forward model. It is difficult to assess the skill of the waveform fitting if e.g. the impact surface roughness changes and snow backscatter on the waveform

shape could be ambiguous.

P7L22ff: The authors should provide information on the detection rates for lead and ice surfaces. In Paul et al 2018 (TC) we needed to introduce different waveform parameter thresholds for Arctic and Antarctic sea ice. There is a risk of introducing freeboard biases if the surface type classification is not performing well (Schwegmann et al. TC 2016).

P7L30: The values for PP in Laxon et al. 2013 are: PP < 18 (not 0.18) and SSD < 4. I assume that the value of 0.18 is only a typo in the manuscript. But the issue that these thresholds are valid for an older version (algorithm baseline-b) of the CryoSat-2 Level-1 waveform. The authors must use baseline-c data for the later years of their data record and waveform oversampling introduced in this version changes pulse peakiness values. The authors should therefore verify their lead and ice detection rates (see above).

P8L8: Same PP threshold magnitude inconsistency. Laxon et al. 2013 reports PP < 9 and SSD > 4 as criteria for sea ice surface returns.

P8L33ff: Does this mean that the authors have removed the DAC and tides from the CryoSat-2 derived elevations and then applied a consistent DAC, tide and mss correction for both ATM and CryoSat-2? What about the inverse barometric correction (see comment above)? This approach however still leaves the ionospheric and tropospheric corrections as an uncertainty factor, which may a dynamic range of 10 cm or more (Ricker et al Remote Sensing, 2016) thus a non-negligible magnitude. A second validation in form of along-track freeboard might help to improve the initial validation.

P9L19ff: Filtering waveform is standard practice. The question is how many are filtered out?

P9L24f: The practice of computing freeboard by subtracting monthly means of sea surface heights and surface elevation is definitely not state-of-the-art. It puts a lot of

trust in the range and tidal corrections that to my experience is not justified. I strongly suggest to estimate freeboard orbit-wise, which also gives a better handle to identify sea surface height estimation issues.

P9L29: The description is a bit confusing, as the earlier sentences reads as snow freeboard (per grid cell) = mean elevation (per grid cell) - mean ssh (per grid cell). From this sentence I get the impression that the authors compute a mean sea surface height, remove that from the all elevations, filter anomalous elevations and then compute the snow freeboard from the remaining values. Please revise.

P9L30: The authors seem to filter negative freeboards within a grid box. I assume this are then from individual waveforms? The filter should be lower than 0 meter or else the negative part of the range noise distribution will be filtered out for thinner ice and thus cause the freeboard to be biased high. In the ESA CCI dataset, the filter range for CryoSat-2 along-track freeboard is -0.25 to 2.25 meters.

P9L34: Is the 125km filter applied to reduce noise, or to interpolate between gaps?

P10L6: The impact of surfaces waves into the ice pack also needs to be considered for CryoSat-2 data.

P10L17ff: For this comparison it would be helpful to have corresponding maps of surface type (lead/ice) detection rates as well as average resnorm values to look into these differences and verify that the sub steps of the CryoSat-2 snow freeboard algorithm are working as intended.

P10L16: Is the ICESat data also filtered to an effective resolution of 125 km?

P10L26ff: It is good that the authors looked into the other output parameters of the waveform fitting. Unfortunately in this shorted form, more question are raised than answered. In essence, the retracking is based on a backscatter model of snow that when applied to CryoSat-2 waveforms, does not result in realistic snow conditions. Again, additional information such as regional differences in surface type detection

rates or waveform fit quality parameters would greatly help to identify potential issues.

P11L34: This is an overstatement given that all existing studies that used CryoSat-2 for Antarctic sea ice are not referenced in this paper.

P19Fig6: axis annotations and ticks are difficult to read on a printed version

---

## Author Response (AR1)

**Author Response to Reviewer Comments**

Dear Editor and Reviewers,

We would like to thank the reviewers for the time they put into reading and commenting on the manuscript. Their remarks and constructive criticism surely helped to improve the quality of this paper. Below you will find our responses to the referee comments, starting with our response to reviewer #1 and followed by our response to reviewer #2 (Stefan Hendricks). The changes made to the manuscript can be found at the end of this document.

Steven Fons and Nathan Kurtz

**Reviewer 1:**

Summary: This paper investigates whether the considerable scattering of satellite radar altimetry at Ku-Band, namely CryoSat-2, that can be expected to occur at the air-snow interface can be exploited to estimate the elevation of the air-snow interface relative to the ocean surface and hence get an estimate of the total (sea ice + snow) freeboard. To do so a two-layer physical model is used together with least square fitting to obtain a fitted waveform to CryoSat-2 Level 1B data from which the elevations are obtained for lat winter / spring October months of 2011-2017. CryoSat-2 elevations are compared with observations from Operation Ice Bridge airborne topographic mapper for two quasi-coincident OIB-CS-2 overflights, one in 2011, one in 2012. Total freeboard is computed and averaged over the entire period 2011-2017 for the entire Antarctic and discussed and compared with ICESat total freeboard maps. Also, the potential to combine air-snow and ice-snow interface elevations for snow depth on sea ice retrieval is tested. While first evaluation results are promising and suggest that total freeboard derived from CS-2 could potentially be used complementary to ICESat and ICESat-2 data more work is required to better understand the observed differences between CS-2 total freeboard and independent data.

I find this an interesting and important contribution to the existing literature and suggest publication of the research results - provided that the authors take into account the various, partly substantial, suggestions for a major revision of their manuscript. I list my major concerns below in the general comments.

We sincerely thank the reviewer for his/her thoughtful and detailed comments on the manuscript. Particularly, we appreciate the suggestions on ways to strengthen the CryoSat-2/Operation IceBridge comparison as well as the encouragement to include more and (more applicable) references to recent studies on Antarctic sea ice freeboard retrievals. Our responses (in blue) to the reviewer comments (in black) can be found below.

General comments: GC1: The introduction lacks to present the state-of-the-art of seaice and/or total freeboard retrieval in the Antarctic. Several studies exist that are using radar or laser altimeter data for this kind of retrieval. In addition, the introduction lacks to present the state-of-the-art of freeboard-to-thickness conversion and inherent problems and uncertainties. While mitigation of the former lack is required to understand why it might make sense to try to derive total freeboard also from CS-2 data, mitigation of the latter lack is required to understand your attempt to retrieve snow depth on sea ice as well.

We address the current snow depths used in thickness calculations, which range from passive microwave data, to assumptions that snow depth = total freeboard, to parameterizations of the snow depth based on multi-year ice fraction and total freeboard. The revised introduction can be found in section 1 of the revised manuscript. The introduction was indeed lacking in the original version of the manuscript. It has been substantially revised to focus on Antarctic retrievals of sea ice freeboard, referencing methods from laser altimetry (Zwally et al., 2008; Kurtz and Markus, 2012; Kern et al., 2016; Li et al., 2018) as well as radar altimetry (Giles et al., 2008; Schwegmann et al., 2016; Paul et al., 2018). Additionally, information on present day freeboard to thickness conversions is included as motivation for retrieving snow depth.

GC2: I don't find the interpretation of Figures 3 and 4 particularly convincing as a motivation why there is substantial(ly more) information about the air-snow interface in the echograms of the snow radar and the Ku-Band altimeter. See my respective specific comments.

These figures are meant to show that Ku-band returns of the air-snow interface are similar to air-snow interface returns from the snow radar. Since the snow radar is utilized often for snow depth retrievals, similar returns from the Ku-band would provide evidence of Ku-band scattering from the air-snow interface and motivation to retrieve the snow surface elevation from CS-2. Responses to the specific comments can be found below.

GC3: Even though Figure 6 and the interpretation is intended to stay qualitative (my guess), I strongly suggest to discuss these results in more detail. Putting more emphasis on increasing the credibility of the elevation estimates at this stage is very important in my eyes. You have the unique opportunity to have quasi-coincident air-borne and space-borne measurements. That's luxury and I have to admit that I am a bit disappointed that you do not exploit this luxury situation further. If I'd be allowed to recommend something, then I would i) quantify the temporal and spatial differences in the tracks and try to investigate whether a correction towards a better spatiotemporal match is worth an effort, ii) collocate the tracks with ice-type information (it might be sufficient to figure out where first-year ice and perennial ice was present), iii) collocate the tracks with meteorological information, e.g. from ERA-Interim or MERRA-2 or perhaps even from one of the higher-resolving weather forecast models to figure out whether air temperatures have been close to 0degC and/or whether and what kind of precipitation potentially occurred (Ideally you have a look at the meteorological conditions of not just the day of the coincident measurements but also of a 1-2 weeks period before to catch potential melt events and hence snow metamorphism near the air-snow interface.). See also my respective comments for Figure 6 and its interpretation.

You are correct that this comparison was meant to stay qualitative and be used simply as motivation for progressing to freeboard retrieval using this method. However, we realize it does provide a great opportunity to evaluate the retrieval and have addressed these points more in the revised manuscript.

We have collocated the tracks with meteorological information from MERRA-2 (both on the day of and 2 weeks leading up to the flight) and ice types on the day of the flight from the EUMETSAT Ocean and Sea Ice Satellite Application Facility (OSI SAF). Both flights took measurements over "first year ice" as noted by the OSI SAF. In the two weeks leading up to the flights, temperatures never rose above 0 deg. C, but did get relatively close (which would potentially indicate some surface melt and change in the surface backscatter). Also on both occasions, there was light snowfall 3-5 days before the flight, with rates staying below 5 mm/day.

The largest noticeable difference between the two dates is a strong north-south surface temperature gradient present during the 2011 flight, while the 2012 flight had near-constant surface temperatures along the flight line, shown in the revised figure 1.

Responses to the specific questions and detailed information on the meteorology leading up to the flight can be found in the revised manuscript and the replies below.

GC4: The interpretation of Figures 7 through 9 would also very much benefit from a more critical discussion which should also involve more work done by other researchers. I find a lack

of attempts to explain the differences observed, e.g. in Figure 9. See my specific comments to these figures.

The results were indeed lacking as far as explaining the differences observed, which could be improved by relating the results to work done by other researchers. To this end, we have added further comparisons with other works, including that of Schwegmann et al. (2016) and Paul et al. (2018) and revised the conclusion section. In reference to figure 9, we agree with your other comments on how this comparison is indirect (i.e. not overlapping in time and involves data coming from different time period lengths) and therefore refrain from giving much explanation of the differences observed or any real quantitative/statistical comparisons. That being said, we have added more explanation to differences observed between other overlapping measurements made from CryoSat-2, found in section 6.2.

Specific comments: Abstract: I suggest to add the standard deviation or Root-Mean-Squared difference in addition to the mean difference values given. If computed, also modal values of the difference would allow to give the obtained values more credibility.

Standard deviation values were added to the abstract. Modal values were computed and included with the figures in text, in Figure 7 and section 6.2, paragraph 1.

Page 1 L30: I guess, since you are focussing on Antarctic sea ice it would not hurt do use citations referring also to the albedo observed over Antarctic sea ice: Brandt RE, Warren SG, Worby AP and Grenfell TC (2005) Surface albedo of the Antarctic sea ice zone. J. Climate, 18(17), 3606–3622 (doi: 10.1175/JCLI3489.1) and Zatko and Warren, Annals of Glaciology 56(69) 2015 doi: 10.3189/2015AoG69A574

Agreed – We have removed the citation for Perovich et al., 2002 (which focused on Arctic sea ice) and replaced it with citations for Brandt et al., 2005 and Zatko and Warren, 2015, found in section 1.

Page 2 Line 2: I suggest to cite Comiso et al., J. Climate, DOI: 10.1175/JCLI-D-16- 0408.1 instead of Beitler 2014; the former is a peer-reviewed paper.

The reference was changed.

Line 6: Please add "Antarctic" or "Southern Ocean" to make clear that these shipbased observation based sea-ice thickness data set is valid there but not general in the polar regions.

Added "in the Southern Ocean".

Line 5-13: - Is there a reason why you refer to multiyear ice only for the Arctic? - Is there a reason why you refer to sea-ice thickness in the Arctic only while for the Antarctic you refer to sea-ice thickness in volume? Is the sea-ice thickness retrieval in the Antarctic more accurate so that it makes sense to also derive the volume?

Multi-year ice was referenced because the Arctic study (Kwok et al., 2009) estimated trends over multi-year ice alone, while in the Antarctic study (Kurtz and Markus 2012) did not discriminate. There was no intentional reason to mention sea ice volume in the Antarctic only, however, this paragraph was revised substantially and no longer includes these references alone (see introduction, GC1 above).

Line 14-20: - I am not sure I like the mentioning of Kwok et al. (2009) and Kurtz and Markus (2012) as the role models for sea-ice thickness measurements from active satellite sensors. Since you are basically referring to the principle, wouldn't it be sufficient to simply write what you wrote without these two references? I I guess my dislike comes from the fact that there have been earlier papers that describe how laser altimetry (which is the main focus in this paragraph) can be used to get an estimate of the total freeboard of snow-covered sea ice: Kwok et al. (2004) or Kwok et al. (2006) for the Arctic and Zwally et al. (2008) for the Antarctic.

The introduction has been revised. Zwally et al. 2008 (among other works) has been included for describing how laser altimetry can be used to retrieve total freeboard (see introduction, GC1 above).

L21-30: - I suggest to re-organize the sentences starting in Line 24 to avoid that sea-ice freeboard is used before being explained. How about you write along these lines: "... 2010-2012. The difference between laser ... [continue until Line 29] ... above the sea surface, known as the "sea-ice freeboard", and is used to calculate sea-ice thickness applying appropriate assumptions (see previous paragraph)."

Thanks for the suggestion – the introduction has been substantially revised and this sections has been changed. The freeboard is defined first, section 1 paragraph 2, and then used after that.

L31-39: - I suggest to expand rightaway in Line 31: "by the depth and variable vertical structure of the snow on top ..." - In Line 32, I suggest to add that it is not simply more precipitation but "... more and more frequent precipitation ..." - Line 34: "sea ice down near the" –> perhaps better: "sea-ice surface down near or even below the"
 - I suggest to break in Line 38 for a new paragraph, starting with "While ... ".

These suggestions have been made in the revised introduction.

- Is there perhaps also the chance that you quantify how large or weak the scattering at the air-snow interface is compared to that at the ice-snow interface? This could make your motivation stronger about why it might be reasonable to look for the snow surface scattering contribution even in Ku-Band.

This explanation is done in section 3 ('Observed Ku-band scattering of radar from Antarctic sea ice'). I've added "(discussed in section 3)" to page 3 line 17 for more clarification.

I strongly suggest to seek for evidence in the literature about the possible strength of the snow surface backscatter at Ku-Band (at nadir) to underline that it is physically reasonable to use CS-2 SIRAL returns for snow freeboard retrieval. I am stressing this because there exists literature in which one aims for snow-depth on sea ice retrieval by confidently assuming that Ku-Band penetrates to the ice-snow interface and combining it with a Ka-Band radar such as from SARAL AltiKa (Guerreiro et al., 2016). Your attempt is clearly conter-acting their assumptions.

We agree that more literature surrounding the strength of the snow surface backscatter at Ku-band needs to be referenced in this section of the manuscript. In particular, we have added Giles et al. (2008), which deals with CS-2 sea ice elevation retrievals in the Antarctic, and Willat et al. (2010), which shows that the strongest Ku-band return can come from either the snow surface, within the snow layer, or the snow-ice interface. More information on the backscatter from the snow surface is found in section 3.

We acknowledge that the dominant backscatter from Ku-band altimetry often occurs within the snow layer on Antarctic sea ice (Schwegmann et al. 2016 and others) and also that it is often approximated to penetrate to the snow-ice interface over Arctic sea ice (Geurreiro et al. 2016, Kurtz et al. 2014, and others). Here, we are introducing the fact that although the dominant scattering occurs below the snow surface, there exists some scattering from the air-snow interface (as shown in Willatt et al. 2010) that we can exploit for snow freeboard retrieval. In that regard, we do not believe we are directly counteracting the assumptions of Guerreiro et al., 2016 and other published works, but instead expanding on the utility of Ku-band returns.

Page 3: Line 10: "builds off" ?
Changed to "builds on".

Line 22: I don't understand the mentioning of the "originally 128". What is this for?
Removed.

Line 34-37: The motivation for choosing data from October is clear. You could have stated why you did not also use data from November. Most of the ICESat spring measurement periods last well into November. Here you state years 2003 to 2009 for ICEsat as years with measurements but actually using you are only data from 2003 to 2007. I can understand that the main motivation for this is to use the data produced by one of you. But from NSIDC and potentially also from University of Hamburg you could possibly have obtained ICESat freeboard data for 2003 through 2009, i.e. from an equally long period as you have CS-2 data from. You stated yourself explicitly, that "Seven years of data allows for a longer-term average to be computed"
We felt that because the comparison between CryoSat-2 and ICESat is (and will always be) indirect, that one month of data would be enough to compare to the spring ICESat campaigns. We agree that both months would be better, but feel one was adequate to assess the distribution. In the later years of the ICESat campaigns, the lower laser energy led to questionable freeboard retrievals, and therefore was not processed using this method. Again, we felt that with the indirect nature of this comparison, the time span chosen was long enough. We have, however, more explicitly stated the differences when discussing the results in the revised manuscript.

Page 4: Line 1: - It would be helpful for a better understanding of Figure 3 to mention the frequency range of the FMCW snow radar. - "First, ..." –> Where is the "Second"?
Added both suggestions to section 2, paragraph 4.

Line 8-12: Is this a gridded product? If yes, which grid resolution does it have and what is done to fill the gaps between the ICEsat overpasses?
Correct, it is gridded. This has been added to section 2, paragraph 5.

Line 13-16: You are using sea-ice concentration data obtained with the NASA-Team algorithm. While when choosing a 50% sea-ice concentration threshold is might not really matter which product to use there is published evidence that the NASA-Team algorithm often severely under-estimates sea-ice concentrations in the Antarctic compared to the truth - particularly in late winter / spring. You could avoid students and early-career scientists being trapped by your choice by choosing a more appropriate sea-ice concentration product right from the beginning, i.e. based on the ComisoBootstrap algorithm or the Eumetsat OSI-450 algorithm.

Thank you for pointing this out. In the revised manuscript, we have used the Comiso Bootstrap algorithm at the 50% threshold. This has been included in the datasets section (section 2, paragraph 6) and in the updated figures.

Line 18: "... altimetry tend ..." –> I suggest to insert: "for ice freeboard retrieval"
Added, thanks for the suggestion.

Line 20: You could cite Willat et al. (2011) here.
Added.

Line 35/36: "This result is expected, as it means that the scattering power from the air-snow interface is closer in magnitude to that of the snow-ice interface in snow radar returns" I do understand your conclusion from the smaller difference in power (about 13 dB for snow radar and 14 dB for Ku-Band altimeter) but I don't understand why this is expected. Lets assume for simplicity that the peak power is 20dB at the icesnow interface for both instruments. Then the power at the air-snow interface would be about 7 dB for the snow radar and 6 dB for the Ku-Band altimeter and with that the power at the air-snow interface would be SMALLER at that frequency from which you assume that the backscattering at the air-snow interface is LARGER. How does this fit together?
What is expected is that the power at the air-snow interface is larger for the snow radar than the ku-band altimeter. This is expected because the snow radar is often used to resolve the two interfaces and calculate snow depth. We do not assume that the air-snow interface power from the Ku-band is larger, but state that because it is similar to the snow radar, that we can expect to have scattering from the air-snow interface in the ku-band.

Page 5: Line 2: At the end of this paragraph interpreting Figures 3 and 4 I have a few questions.
i) How accurate are the two instruments with respect to the dB values shown?
The two instruments aren't radiometrically calibrated, so it is difficult to know the accuracy with respect to the dB values shown (Some radar parameters from each can be found in Panzer et al. 2013 (snow radar) and Gomez-Garcia et al. 2012 (Ku-band, conference paper) ).
Along the flight lines presented here, the snow radar had a mean noise level of -29.1 dB (std:1.39 db) and a mean air-snow interface signal level of -16.8dB (std: 1.47). The Ku-band had a mean noise level of -30.3 (std:1.32 ) and a mean air-snow interface signal level of -17.9 (std: 2.09), showing that the return from the air-snow interface is well above the noise level in both cases. We have included these values in the caption of figure 3.

ii) How relevant is the similarity in the histograms shown in Figure 4 with respect to the shared mode at about 11dB while at the same time the histogram shows secondary modes at 16dB (snow radar) and 20 dB (Ku-Band).
The point of this figure was to show that the histograms are indeed similar to strengthen our case for tracking the air-snow interface from Ku-band. The shared mode is encouraging in that regard, while the secondary peaks could mean that the snow radar is more sensitive to resolving both interfaces when the power difference is larger between the two (as is expected, since the snow radar is used for deriving snow depth).

In addition I have a few comments: iii) What were the meteorological conditions during that flight? Can we expect homogeneous snow properties in terms of snow wetness etc.
iv) How would Figures 3 and 4 look for the October 2011 campaign? Would they result in the same result?
We would expect relatively homogenous conditions during the 2012 flight, judging from the surface temperatures (from MERRA-2, below) and the ice type (taken from EUMETSAT Ocean and Sea Ice Satellite Application Facility) which gives the same "first year ice" classification everywhere along the line.

[Figure]

In 2011, the ice type was again classified as first year ice everywhere along the line, however, there was a stronger surface temperature gradient along the flight line (shown below). This gradient may have resulted in inhomogeneous snow properties that could have altered the radar returns. In this regard, figures 3 and 4 in the 2011 campaign may have looked slightly different than the 2012 campaign and could be responsible for features observed in figure 6.

[Figure]

MERRA−2 Surface Temperature
19:00UTC − 10/13/2011

Temperature (Deg. C)

−25.0 −22.5 −20.0 −17.5 −15.0 −12.5 −10.0 −7.5 −5.0 −2.5 0.0 2.5 5.0

The meteorological conditions during these flights have been added to figure 1 and discussed (along with the ice types) in the revised manuscript, under the initial validation (Section 5).

v) What is the length (in kilometers) of the transect (or echogram) shown in Figure 3?
The echogram here is taken from a single datafile, which in this case covers just over 3km (3.016km). This has been added to the figure description.

vi) The Ku-Band altimeter histogram in Figure 4 has a substantially longer right tail with high dB values. It is almost certain that these values are responsible for the 1dB difference observed between the snow radar and the Ku-Band altimeter data. Are these particularly large differences the result of a particularly low power at the air-snow interface compared to the peak power or is this the result of peak powers being generally elevated at Ku-Band compared to the snow radar?
Agreed – the tail is likely responsible for the difference between the two observed. The differences between the histograms are probably a combination of both, since the snow radar generally has higher power at the air-snow interface (as it's used for snow depth retrievals) and also since peak Ku-band returns tend to be larger than peak snow radar. However, this figure was just meant to highlight the similarity in the histograms, which we feel it adequately does.

vii) What explains the larger time difference between the locations denoted by red and black points in Figure 3 for snow radar echogram compared to Ku-Band?
The snow radar tends to resolve the interfaces better, as can be seen in the more separate "interfaces" tracked by the peak-picker. The Ku-band, at a higher frequency, tends to be slightly less sensitive to the air-snow interface and also attenuated more by the snow volume, and thus has less-defined interfaces and smaller time differences.

viii) What is the source for the staggered echos above the air-snow interface at Ku-Band? Such echos are not at all present for the snow radar data.

These staggered echoes are returns from radar sidelobes, which are primarily a function of non-linearities in the transmit pulse. Over strong ice returns (seen in the figure) sidelobes can be present in both radar echograms, however in this case, the Hanning time-domain window filter applied to the data removed sidelobes from the snow radar but not the Ku-Band. Our peak-picker is designed to ignore staggered echoes such as these.

Line 21: What is "n"?
The variable "n" is part of the summation ($\sum_{n=0}^{N_b}$ ) and stands for each of the synthetic beams ($N_b$) in the sum.

Page 6: Line 2: I guess "thicker snow depth" and "scattering effects from the snow surface" are not as much linked with each other as scattering effects from the snow volume. What makes a thicker snow cover to have more surface scattering than a thinner snow cover?
This line was intended to mean that the entire snow layer (encompassing both the surface and volume) cannot be neglected. It has been rephrased to read "…scattering from the snow layer cannot be neglected…" to reduce confusion, section 4.1.

Line 27: Why "Though"?
Removed "though" and added "however" in the last paragraph of section 4.1, for clarity.

Page 7: Line 16: What is the motivation to only use a different initial guess for alpha in case resnorm is too high after the first fitting attempt?
The motivation to only use alpha comes from the fact that it is the least well-constrained initial guess variable, and that re-fitting with a new alpha tended to reduce the fitting error. While other initial guesses could have been modified and re-fit, we felt that it was not necessary due to the facts that a) they are all empirically derived and b) the added computation cost did not outweigh the benefit.

Page 8: Line 8: Are these PP and SSD thresholds also taken from Laxon et al. (2013)?
Yes, the PP and SSD thresholds are taken from Laxon et al., (2013) though there is scaling factor of 100 difference present in the PP. As the PP was not explicitly defined in Laxon et al., 2013 the values are calculated following from Armitage and Davidson (2014), and SSD comes from the CryoSat-2 product. These references have been added.

Line 12: "seasonal average freeboard datasets" –> perhaps better "datasets of the seasonal average total freeboard"?
Changed.

Line 15: A very good place to cite the paper by Kwok and Maksym, 2014 in J. Geophys. Res., doi:10.1002/2014JC009943
Added.

Lines 23-25: Please explain how the OIB data area used. Did you take data from both flights? Did you average over all valid points? What is meant by "respective surfaces"? Two final question at the end of this section: What happens in the special case of a snow free ice floe?

What happens in case of a wet snow surface, where penetration of the Ku-Band into the snow cover is almost zero?

Added some clarification to this paragraph. Data were taken from the Ku-band radar from both flights, and the values found were the average of all valid points. The "respective surfaces" refer to the surface backscatter coefficients of the air-snow and snow-ice interfaces.

As far as the cases you mentioned: From CryoSat-2, it depends on the return waveform parameters (PP and SSD). In both of these cases, the return may be more specular than a typical snow-covered floe and be non-classified or classified as a lead. The chosen thresholds tend to limit misclassifications, but as with any retrieval process, misclassifications can still occur. We have added some more verification figures (5 and 6) to show how the algorithm performs over different regions and surface types.

Line 32: "found" –> perhaps better "computed" or "derived"?

Agreed, it has been changed to "derived" in section 5 paragraph 1.

Page 9: Lines 7-13 & Figure 6: - Please provide a measure of the total distance along the tracks shown in Figure 6 a) and b). This would make referencing to certain feature more easy in addition to simply providing an easier interpretation of the spatial scale we are looking at. - I suggest to add a vertical line at zero difference ATM minus CS-2 in images c) and d). - What is the average difference in successive measurements in images a) and b)? - Suggest to use the same y-axis scaling for a) and b) for a better visual comparability of the elevation variations.

The suggested changes have been made to figure 6. Instead of showing the lat/lon for each X axis tick, only along track distance is shown to reduce clutter and improve visibility. The starting and ending lat/lon pairs are included in the figure caption.

The distance between successive measurements is 0.38km (corresponding to the along-track footprint of CryoSat-2).

- I'd say that the overall agreement, i.e. the large-scale agreement is better for 2011 than 2012. - For 2011 the mean is very close to zero, right. But the modal value is between 5 and 10 cm with CS-2 underestimating ATM elevation. - CS-2 elevations quite often exhibit strong variations in magnitude; in 2011 more during the first third of the track, in 2012 during the first two thirds of the track. These strong variations (or jumps) are as large as about 40 cm and except in one case do not have a counterpart in the ATM elevations. Please try to give an explanation to these.

We believe these jumps are a product of the threshold chosen to represent "good fits" in the retrieval process. Currently, we use a residual value ('resnorm' in text) of 0.3 as the cut off for acceptable waveform fits. Fit waveforms with larger residuals are not included in the retrieval. When they are included, we see more points with anomalous elevation (i.e. more jumps). Lowering the threshold below 0.3 does help to remove some of the jumps, but also removes non-anomalous points and results in worse overall agreement between CS-2 and ATM. Therefore, we still find 0.3 to be the most appropriate threshold, and understand that the jumps are caused by "bad" fits that are still within our threshold of being "good". An explanation has been added to the text, section 5 paragraph 3.

- While in 2011 the large-scale agreement is quite good (you could even stress this impression by adding elevation profiles with largescale smoothing applied), in 2012 there appears to be a

systematic under-estimation of the ATM elevation by CS-2. Please try to give an explanation to these as well.

This potentially comes from the smaller footprint size of ATM compared to CS-2. ATM is able to resolve small-scale peaks that would get washed out in the CS-2 return. While it is not as apparent in 2011, there still exists a slight underestimation of ATM by CS-2 in the last third of the profile. This potential explanation has been added to section 5, paragraph 3.

- You argue that differences between the two elevation data sets might be caused by "initial temporal and spatial discrepancies between the two data sets. Would you be able to quantify these differences? Would it make sense to do a correction of the track of one sensor with respect to the track of the other sensor?

The discrepancies mentioned here mainly refer to the sampling differences between the two instruments, which would be difficult to quantify. The CS-2 footprint is around 1.6km across track and 0.38 km along track, while the ATM icessn footprint is much smaller (~250m across and 30m along). This itself would lead to different ice being sampled between the two instruments, and very likely different mean elevations per shot.

From this initial validation effort, we felt that the agreement was good enough to justify a freeboard calculation, and that performing a correction (for drift or footprint size) wouldn't factor into the freeboard retrieval, and therefore did not include it in this study.

- You write that both datasets "appear to detect similar locations of troughs and ridges along the flight line". I don't find this statement particularly convincing because there are also many cases where troughs in one dataset and ridges in the other dataset coincide.

Agreed. This statement has been removed from the text.

Lines 19-33: - Line 25: "fewer than five data points" –> What is the distance of successive data points? How many data points would fall into one 25 km grid box at maximum, i.e. diagonal crossing? Can you comment on the data density as well? How many CS-2 overpasses or days with CS-2 overpasses in one grid cell do you have in one month?

Each successive CS-2 point is .38km, which would result in a potential maximum of about 93 data points per orbit. The number of data points in each grid cell varies greatly depending on month and location relative to the continent. Values range typically from 0 (furthest north) to over 200 (further south, with multiple orbit passes over certain cells) raw data points. After filtering and removing anomalous freeboard measurements, around 58% of the initial waveforms remain. We have added figure 5, which shows this valid waveform fraction on a pan-Antarctic scale.

Lines 29/30: "Any points within each grid box ..." –> Could it be that this sentence should be placed before the previous sentence? I am asking because the previous sentence already describes the method used at grid level.

We are updating our freeboard calculation following comments from the other reviewer. This sentence has been removed.

– Line 32: "are smoothed" –> Why is this? Why do you do that? Is it because of the gaps between the overpasses? Please state so in the paper.

This is done mainly to reduce noise but also to fill gaps in the data. There are some gaps in the data between the overpasses and in grid cells that have been filtered out.
The following sentence has been added to the end of the paragraph: "Smoothing is applied to reduce noise in the CryoSat-2 data and also to fill in gaps in the data." (section 6.1, paragraph 1).

Page 10 Paragraph ending in Line 7: - While there is not too much work yet about freeboard distribution from radar altimetry in the Antarctic I still suggest that you consider comparing your results with the results published by Giles et al. (Geophys. Res. Lett., 2008), Schwegmann et al. (Annals of Glaciology, 2015), and Paul et al. (TC, 2018). - While the work of Nghiem et al. (2016) is really interesting and certainly not invalid over parts of the Antarctic MIZ I suggest that you also take into account (and at least mention if not discuss) the potential effect of ocean swell, lower CS-2 data density and hence a larger representativity error, and ice types being different in the MIZ than in the pack ice; a large fraction of the Antarctic MIZ is formed by the often several hundreds of kilometers of pancake ice or cake ice or first-year ice with small floe sizes (< 100 m) for which I doubt that CS-2 is going to provide reasonable elevation and hence freeboard estimates.
We have added many references that were lacking in the original manuscript, and our results have also been compared qualitatively to the freeboard distributions shown in Schwegmann et al. (2016) and Paul et al. (2018), in section 6.2.
Additionally, we have expanded our explanation of the ice edge in the revised manuscript to include effects of ocean swells and surface waves as well as the lower CS2 data density and differing ice types in the MIZ. This can be found in section 6.1, paragraph 2.
Figure 5 shows valid waveform fractions (as well as lead /floe fractions). Over the MIZ, there is a higher number of invalid waveforms (speaking to the complex ice types found here) that are not included in the freeboard calculations.

- I note that the distribution of total freeboard shown in Figure 7 is quite patchy and contains several artificial south-north oriented freeboard variations (possibly caused by sampling issues). I note that the freeboard in the southern Ross Sea is indeed lower than further north. However, given the fact that this is an area of extensive new-ice formation and export paired with low precipitation and hence thin snow cover, the freeboard values shown are certainly at the higher end of what is typical there - if not a proper overestimation. Sea-ice thicknesses in the southern Ross Sea are 20 to 50 cm ... total freeboards (without snow) therefore in the range of between 2 and 5 cm and not between 10 and 20 cm as indicated in the maps.
We have updated our freeboard calculations (to include on orbit filtering, at the request of the other reviewer) and thus the distribution differs slightly in the areas mentioned. However, the total freeboard is heavily dependent on the snow depth, which (even if it's small) is unknown. Also, these total freeboards in the Ross Sea do compare reasonable well with that found using ICESat in this region, and are similar to what is shown in Schwegmann et al. (2016) and Paul et al. (2018). Since there is no or little snow, the total freeboard should be close to the radar/ice freeboards. The added figure 5 shows that more lead-type waveforms are found in the Ross Sea, meaning fewer floe points are used to calculate freeboard than in other areas, which could explain the slightly higher-than-average freeboard values shown here.

Lines 8-15 & Figure 8 - "smallest measured freeboard" –> perhaps better "smallest measured mean October freeboard"

Changed.

- 25.77, 27.6, 12.97 ... I suggest to give these figures with the same number of digits, i.e. 25.8, 27.6 and 13.0.
Changed.

- Showing the mean total freeboard together with the sea-ice area is certainly fine even though, as you stated correctly, it is not too clear why you find a good correlation between these two quantities. However, instead of the sea-ice area one could plot other variables as well. One would be the standard deviation of the mean total freeboard as a measure of the scatter of the mean values. A second one would be to show either the number of 25 km grid resolution grid cells with valid CS-2 data or even the number of individual valid freeboard (or elevation) measurements. Since you have many gaps in the original CS2 data it would potentially be a very interesting additional information. - I note that the maximum inter-annual difference of the mean October freeboard is 2 cm. Is this within or outside the retrieval uncertainty?
We definitely agree that we could plot different variables here, however, the point of the figure was not meant to be further validation of the algorithm, but instead to simply show that a relationship may exist and could be explored further. We have added additional figures / information throughout the manuscript that act to further validate the performance of the algorithm. More coincident measurements are needed to fully quantify the uncertainty in this method, something that will be done in future work.

Lines 17-25 and Figure 9: - I suggest to color the open water in a grey tone (different than Antarctica of course) to ease discrimination between areas with differences close to zero and open water. - I suggest to reduce the range of the differences shown to +/- 30 cm to show more details. The way the range is chosen currently only reflects the larger differences.
We have revised this figure to account for these issues, but instead changed the color bar to include grey instead of white (Figure 10).

 - I find it essential that you mention three things in your discussion of this Figure: i) the larger number of years for CS-2 (7 instead of 5), ii) the fact that the ICESat data cover different time periods with at least 2 of the five years have a substantial if not dominating overlap in time with November and hence conditions changed towards spring (As far as I recall you analysis you did make the effort the average CS-2 from exact those dates from which also ICESat measurements exist.), iii) the fact that we look at years 2011-2017 for CS-2 but 2003-2007 for ICESat, i.e. two different, not overlapping time periods. While this might not have an effect it needs to be stressed once more in the context of this discussion. Finally, iv) one could ask whether you used the same method for averaging the CS-2 data (and filling gaps, extra- or interpolating gaps) than was done in Kurtz and Markus, 2012? Because of these four issues I warmly recommend to delete the last sentence in Lines 24/25.
We tried to emphasize that these were not direct comparisons, but agree that more could be added to fully explain the differences between the two datasets. We used the same method for averaging and filling gaps as was used in Kurtz and Markus (2012), however with an updated freeboard calculation, the averaging has been done slightly different in the revised manuscript, though the gap filling has remained consistent.

These points have been added to the description as its own paragraph (section 6.2, paragraph 2) and the last sentence has been removed.

- When talking about a difference of only 1.9 cm: What are the uncertainties in monthly mean freeboard from CS-2 and from ICESat? Is the difference about the uncertainty?
The estimated uncertainty in the ICESat freeboard is 1.8cm (Kurtz and Markus, 2012). Unfortunately, freeboard from the IceBridge underflights in the Antarctic have not yet been processed by the OIB team, and therefore no coincident "true" values exist from which to calculate uncertainty. Future work will compare freeboard calculations directly to better characterize the uncertainty in this method.

- You do not make any attempt to explain the highlighted negative freeboard differences CS-2 minus ICESat in the Weddell and Amundsen Seas. Why?
This has been inadvertently left out of the original manuscript. It could be due to a number of reasons, however, since the time periods of data from ICESat and CryoSat-2 do not overlap, differences observed aren't necessarily due to discrepancies in retrievals (i.e. both methods could both be showing the "true" freeboards). Therefore, we tried to refrain from explaining differences from ICESat. We add some explanation of differences seen between our method and the works of Schwegmann et al. (2016) and Paul et al. (2018), found in section 6.2, paragraph 3.

- How do your results compare to the work of independent researchers: Yi et al., 2011, Kern and Spreen, 2015, Kern et al., 2016, Li et al., 2018?
Instead of comparing our results to more studies that use ICESat data, we have instead elected to compare our results to studies using CryoSat-2, to see how the freeboard distribution compares to other radar-based products. Specifically, we have qualitatively compared our results to Schwegmann et al. (2016) and Paul et al. (2018), found in section 6.2.

Lines 27-36, Figure 10: - Figure 10 is indeed quite interesting because the highest "snow depths" are not observed in the Weddell Sea but on East Antarctic sea ice. Puzzling. This is even contradicting your own work (Kurtz and Markus, 2012), where the freeboard maps shown are assumed to represent the snow depth while assuming sea-ice freeboard to be zero. In that work maximum freeboard and hence "snow depth" was observed in the Weddell Sea. What is further interesting is that the "snow depth" is nowhere considerabler smaller than 10 cm - even not in the southern Ross Sea where there is little or no snow on the young sea ice. - I would have found it again very useful, if you would have related the results shown in Figure 10 also with other work, i.e. snow depth based on satellite microwave radiometry (Markus et al., various) or based on ICESat data (Kern and Ozsoy-Cicek, 2016).
We didn't compare these retrieved "snow depths" to other sources, since we believe the snow-ice interface is not being tracked well in this algorithm, and thus these results do not accurately represent the actual snow depth distribution. Further work will need to be done to improve the retracking of the snow-ice interface to retrieve snow depth, if indeed it is possible given the attenuation of the radar signal in flooded/briny snow.

Page 11: Lines 1-4: That the peak-picking algorithm provides snow depth along that flight line which is within 10% of the values published by Kwok and Maksym (2014) is an encouraging result and should be highlighted more.

We agree that it is indeed encouraging, but have done little in the way of close verification of the interface detection for each data file along the flight path. We feel it is best to refrain from highlighting this result until more validation is done.

Lines 4-6: I am pretty sure that this is a frequency issue and not an issue of bandwidth or footprint size: The snow radar used on OIB is a 2-8 GHz radar, right?, while CS-2 operates in Ku-Band.
With the "correct" snow conditions it is very likely that CS-2 does not penetrate down to the ice-snow interface, explaining the considerably lower "snow depth" value estimate from the two elevations. Actually, the OIB snow depths are potentially even higher because of the difficulties to retrieve snow depth in areas of deformed sea ice and on multiyearlike ice reported elsewhere. You're correct that CS-2 likely does not penetrate down to the snow-ice interface, and this is something we included in explanations in the previous paragraph. Assuming that CS-2 is tracking the correct snow-ice interface (as other works have done) we would still expect the snow radar-derived snow depth to be larger due to the footprint size and bandwidth discrepancies between the two instruments.

Comment: Again I doubt that the precision and accuracy of the data warrants to give mean values with 3 digits = millimeter precision here. I guess 0.29 m, 0.26 m and 0.15 m would do it.
Changed.

Line 8: I suggest to delete "slightly". It is considerably greater.
Removed "slightly".

Line 12: "validating" –> perhaps better "evaluating" or even only "understanding".
Changed to "evaluating".

Line 13: "to better understand the snow depth distribution on sea ice" –> perhaps better: "to use it together with the air-snow interface for snow depth on sea ice estimation."
Changed.

Lines 15-27: - Line 16: "air-snow interface of sea ice" –> perhaps better "air-snow interface of snow on sea ice".
Included "snow on".

- Line 20: "validate" –> "evaluate"
Changed.

- Line 24: "data from comes from" ...?
Removed the first instance of "from".

- Line 22-27: I strongly suggest you revise these conclusions based on the additional analysis, interpretation and discussion that is recommended in the general comments. In particular, figures for standard deviation and potentially also uncertainties should be given in addition to the mean values. One can have a mean value close to zero with one part of the data pairs having -50 cm difference and the other part having +50 cm difference ...

The conclusions have been revised based on the changes mentioned in these comments, especially adding the standard deviation values from these differences and reflect the updated freeboard calculations, figures, and comparisons to other sources.

Lines 28-33: - Line 28: "retrieved ice freeboard" –> you did not really retrieve ice freeboard, did you? You computed the snow depth from the difference between air-snow interface elevation and snow-ice interface elevation.
Correct. This was changed to refer to the retrieved snow-ice interface elevation values.

- Line 29: "lower than typically expected" –> since you did not show any other results about snow depth on Antarctic sea ice - expect for the case of East Antarctic sea ice - it is difficult to follow this statement.
The "lower than typically expected" was broadly referring to other studies that measured snow depth on sea ice, though a citation was missing from this manuscript. The revised manuscript includes a reference to snow depth measurements made from passive instruments (Markus and Cavalieri, 1998) and in situ surveys (Massom et al., 2001).

– Line 31: I agree about the potentially wide-spread flooding of Antarctic sea ice but sea ice is mostly flooded when the ice-snow interface is submerged be low the sea level and in that case an ice-snow interface does not exist in that sense anymore. If it still exists, e.g. through to lateral flooding it is possibly close to zero. It might therefore be more correct to again refer to sea water and brine wicked up into the snow, creating a saline snow - non-saline snow interface which is possibly the interface seen by Ku-Band.
Agreed, we have revised this section to mention the brine layer that is likely detected in the Ku-band (section 7, paragraph 1).

Question: For the location of the air-snow interface you have a-priori information from seasonal mean ICESat snow freeboard. For the ice-snow interface you don't have any a priori information, do you? This could be one explanation for the sub-optimal performance with regard to detecting the ice-snow interface as well.
This could be one explanation for the sub-optimal performance, yes. Another could be the fact that although the threshold retracking used here has shown to be successful at retrieving the snow-ice interface in other works (Laxon et al., 2013; Kurtz et al., 2014) the addition of more free parameters could lessen the usefulness of the threshold chosen for snow-ice interface tracking.

- Line 34: I guess it would be fair to cite the already existing literature about using CS-2 data for Antarctic freeboard (sea-ice thickness) retrieval: Paul et al., 2018, and change "... observing Antarctic sea ice." into something like: "... observing Antarctic sea ice with satellite radar altimetry in addition to Paul et al. (2018)." Maybe there is even more work out using CS-2 data in the Antarctic? Please check!
We have revised the conclusions to include the Antarctic freeboard retrievals made by other authors (Schwegmann et al., 2016; Paul et al., 2018), and focused on the novelty of retrieving snow freeboard (as opposed to ice freeboard) from CS2.

Page 12, Line 7: Again I think it would not be too bad to add the work of other authors here to avoid the impression that you are the first on this field: "... for improved retrievals of Antarctic sea ice thickness, complementary to sea-ice thickness retrievals based on the 15+ years long time series of combined Envisat - CryoSat-2 freeboard estimates (Paul et al., 2018)."

We have included the work done by ESA's CCI Sea Ice group (Schwegmann et al., 2016; Paul et al., 2018) in this statement, to exemplify how long time series of sea ice freeboard and thickness can improve understanding of sea ice overall.

Page 23: Line 23: Giles et al. This paper was in Geophys. Res. Lett. and not The Cryosphere

Changed – thank you.

**Reviewer 2 (Stefan Hendricks):**

In their paper "Retrieval of snow freeboard of Antarctic sea ice using waveform fitting of CryoSat-2 returns", the authors develop and apply an algorithm to obtain snow freeboard of Antarctic sea ice using waveform fitting of CryoSat-2 data. The waveform fitting is based on a forward model from earlier work of one of the authors and the main work here has been the to include backscatter from multiple interfaces (air/snow and snow/ice) in combination with snow volume backscatter for the application of sea ice in the southern hemisphere with its higher and more complex snow load. The authors compare the results with airborne validation data and earlier ICESat laser altimeter results and conclude that their algorithm can be used to obtain snow freeboard with the CryoSat-2 during the maximum austral sea ice extent in October. They also investigate the potential to retrieve snow depth from CryoSat-2 waveforms, but do not find realistic results except for an area in the East Antarctic sector.

I have been part of the team that produced freeboard maps of Antarctic sea ice from Envisat & CryoSat-2 data in the Climate Change Initiative. We used an empirical retracking scheme, which made it difficult to include the contribution of snow backscatter in the freeboard algorithm without prior knowledge of its impact on waveform shape. We are acutely aware of this deficiency in the CCI sea ice thicknesses for the southern hemisphere, and it is commendable that the authors attempt to overcome this issue. Therefore this study is for me a very welcome and novel contribution to the field of remote sensing of Antarctic sea ice thickness. The concept is generally sound, but there are a number points that need to be addressed before publication. I have detailed my concerns in the general and specific comments below:

We would like to sincerely thank the reviewer, Stefan, for his important and insightful comments on this manuscript. The insight he shared from his own experience with CCI Antarctic sea ice freeboard retrievals certainly helped to improve the quality of this retrieval method. Our responses (in blue) to each of the comments (in black) can be found below.

General Comments:
1) It is understandable that maps of snow freeboard is the main objective of this work, but the authors show very little in terms verification of the different algorithm steps. The waveform fitting is based on nine free parameters, but it is not clear to me what the sensitivities are for resolving the snow backscatter properties. E.g. do snow backscatter/depth and surface roughness have an ambiguous impact on waveform shape and thus range? In general, the potential impact of surface roughness changes receives very little attention in the evaluation of the results. The fact that the snow depth resulting from the waveform fitting is unrealistic in wide parts of the study regions shows that there are issues with fully resolving the backscatter processes. A sensitivity study of the waveform forward model would help greatly to assess the skill of the algorithm for different snow conditions. Regional information on the average waveform fit quality would also be of interest to the reader as it might help to identify issues of the algorithm. I find this especially important as the direct validation with the

Operation IceBridge ATM and snow radar data is very limited compared to the spatial and temporal extent of the CryoSat-2 data and the otherwise indirect comparison to ICESat.

We realize the original manuscript was indeed lacking in the verification of the sub-steps of this retrieval method, and have added additional figures and information throughout the revised manuscript.

The impact of surface roughness on the results was not discussed because we felt the impact would be small for our objective, which is simply to show the possibility of this technique to derive snow freeboard. Additionally, surface roughness is not explicitly handled by the model, but instead, backscatter values are taken from an initial guess and derived through the waveform fitting process. We acknowledge that surface roughness does play an important role on the backscatter properties, but it is one that will be explored in future work aiming to improve the accuracy of this algorithm. The revised manuscript will mention this fact. Waveform fit quality will also be made apparent in the revised manuscript, focusing on goodness of fit distributions across the sea ice pack as well as a discussion of the percentages of waveforms that are filtered out in this process (revised figure 5, below). Surface type classification fractions will also be included to better assess the performance of the algorithm. See specific comments below for more details.

2) The conversion of surface elevations into snow freeboard is not state-of-the-art, in parts problematic and the authors risk to undermine the value of their retracker development work. For example, the authors use a surface type classification scheme that was originally designed for Arctic sea ice and an earlier version (baseline-b) of the CryoSat-2 Level-1 data. For the ESA CCI data set we had to define separate waveform parameter threshold for Arctic and Antarctic sea ice. The authors need to show that there is no preferential sampling of surfaces that may introduce a freeboard bias, which can be easily done by providing information on detection rates for lead and floe surfaces and compare those the surface type fractions of the ESA CCI dataset (Paul et al. TC 2018). I would also strongly suggest to compute freeboard per orbit and not by subtracting monthly mean elevations and sea surface height. In our experience the geophysical range corrections in the CryoSat-2 Level-1 product files are not good enough for this approach. Using the instantaneous sea surface height during the orbit will be more reliable and also yield better options for filtering incorrect retrievals.

It is acknowledged that the lead/floe classification scheme used here was originally developed for the Arctic which deserves further analysis, however since our methodology maintains the same 128 bin sampling as the Baseline B data there should be no difference with regard to product version. In the added verification steps, we have shown detection rates for the lead/floe classification scheme, shown in the revised figure 5:

[Figure]

There are some differences from Paul et al. (2018); here, there appears to be a smaller sea-ice-type waveform fraction overall and specifically a greater concentration of lead points in the Ross Sea as compared to Paul et al. (2018). It is likely that this method is classifying smooth, new ice in the Ross sea as lead points – a fact that is discussed in the revised manuscript. The maps above are from October 2016 while Paul et al. (2018) has maps from September 2011, which could be a potential source of the differences observed. A more detailed discussion has been included in the revised manuscript, Section 4.3.
In addition, we have updated our calculations of freeboard slightly to include on-orbit filtering, which is explained below and in the revised manuscript, section 6.1.

3) Several earlier studies and datasets that are highly relevant for this topic are not mentioned and the manuscript gives the wrong impression that the work of the authors is the first application of CryoSat-2 for Antarctic sea ice. The authors also state that the impact of snow backscatter on ranging with Ku-Band frequencies is "often ignored" which I certainly do not agree with. It might be a matter of wording, since most operational CryoSat-2 products use the assumption that the freeboard is the average ice freeboard within the footprint. But this is due to the challenges of parametrizing snow backscatter and its temporal and regional variability, not the lack of awareness in the scientific literature. I have suggested references in the specific comments below. The

lack of reference to existing publications that specifically deal with CryoSat-2 freeboard retrieval in the southern hemisphere (Schwegmann et al, 2016, Paul et al, 2018) is also unfortunate, as these are a good motivation for this study. There would be added value if the authors compare their snow freeboards to the freeboard information in the ESA CCI CryoSat-2 data set (see data availability in Paul et al. TC 2018) and demonstrate the improvements and limitations of using waveform fitting.

We agree that the original manuscript was lacking in terms of references to published literature on the topic. The introduction has been completely revised to include recent applications of satellite altimetry to Antarctic sea ice, referencing Giles et al. (2008), Schwegmann et al (2016), Paul et al. (2018), and others. We have also revised other parts of the manuscript to make it more clear that the state of the science is well aware of the impact of snow backscatter on ranging from Ku-band and is not simply ignoring this fact. A comparison of the retrieved freeboard to that from Paul et al. (2018) would indeed be useful and has been included in the revised manuscript, in section 6.2. Additionally, we have revised the conclusions (section 8) to make it more clear that this is not the first application of CryoSat-2 for Antarctic sea ice, and instead stressed the novelty of retrieving snow freeboard from CryoSat-2.

Specific Comments:
P2L7: I guess you mean "active" in the sense of active microwave sensors respective altimeters in general? I would recommend to use the term "satellite altimeters" instead of "active platforms" throughout the document. (Typo: remove -> remote)

The "active" was intended to be in contrast to the "passive instruments" mentioned in the previous paragraph, but admittedly added some confusion. All mentions of "Active" instruments and platforms have been changed to "satellite altimeters".

P2L27: Typo: "of off"
Corrected to just "off".

P2L28: Beaven et al. 1995 states that the snow/ice interface is the dominated backscatter source. To my knowledge Beaven itself does not imply that cold snow under laboratory conditions is completely transparent for Ku-Band. This is a fine distinction but relevant for this paper.

Agreed, this is a necessary distinction to make for this work. The intro has been revised, and the new sentence reads "Most radar altimeters operate in the Ku band at around 13.6 GHz, a frequency that has been shown to produce a dominant backscatter from the snow-ice interface (Beaven et al., 1995)."

P2L39: Please rephrase the term "often ignored" as it does not properly reflect the state of the scientific literature. The issue is not lacking awareness of the importance of snow interface or volume backscatter for Ku-Band freeboard retrieval (e.g. Armitage and Ridout GRL 2015, Ricker et al. GRL 2015; Nandan et al. GRL 2017), it is the challenge of getting the temporal and spatial variability from the available waveforms. A better description would be "often not included freeboard retrieval algorithms, especially those depending on an empirical waveform evaluation".

That is better, thank you. We have revised this line to match the suggestion and included the appropriate references.

P3L30: The inverse barometric correction is included in the dynamic atmosphere correction. Both corrections should not be applied in combination.
Thanks for pointing this out, it was an error in the text as only the dynamic atmospheric correction is applied. For accuracy/clarity, the sentence has been changed to read "Geophysical corrections are applied by using the CryoSat-2 data products, which include the ionospheric delay, dry and wet tropospheric delay, oscillator drift, dynamic atmosphere correction (which includes the inverse barometer effect), pole tide, load tide, solid Earth tide, ocean equilibrium tide, and long period ocean tide."

P4L18: See comment above. There are several publications that investigate the impact of snow on CryoSat-2 freeboards and are not cited here.
We have added citations to a number of works that were missing from the original manuscript, and acknowledged recent studies dedicated to investigating the effects of the snow layer on Ku-band ranging and freeboard retrievals to the beginning of this paragraph. This includes references to Armitage and Ridout (2015); Ricker et al. (2015); and Nandan et al. (2017).

P7L7ff: At this point it would have been good if the authors had included a sensitivity study using their forward model. It is difficult to assess the skill of the waveform fitting if e.g. the impact surface roughness changes and snow backscatter on the waveform shape could be ambiguous.
Agreed – here we added a study (Figure 6) looking at the initial guesses for the standard deviation of surface height and the total backscatter, showing modeled waveforms, fit waveforms, and the corresponding freeboard maps (shown as a difference from the parameters chosen in the rest of the study).
It is important to note that surface roughness is not explicitly included in the current form of the model, as we feel the impact of surface roughness will not affect our goal to show that snow freeboard retrievals are possible with CS-2. Surface roughness changes are important and will surely need to be taken into account in order improve the accuracy of the retrieval method, but this is something that will be done in future work.

P7L22ff: The authors should provide information on the detection rates for lead and ice surfaces. In Paul et al 2018 (TC) we needed to introduce different waveform parameter thresholds for Arctic and Antarctic sea ice. There is a risk of introducing freeboard biases if the surface type classification is not performing well (Schwegmann et al. TC 2016).
Thanks - we have included this in figure 5. Overall, the distribution of leads and floes is generally comparable to that found in Paul et al. (2018), though differences are discussed in the text. Additionally, we have included the percent of valid waveforms, as well as the average resnorm value over the study period (October 2011-2017).

P7L30: The values for PP in Laxon et al. 2013 are: PP < 18 (not 0.18) and SSD <

4. I assume that the value of 0.18 is only a typo in the manuscript. But the issue that these thresholds are valid for an older version (algorithm baseline-b) of the CryoSat-2 Level-1 waveform. The authors must use baseline-c data for the later years of their data record and waveform oversampling introduced in this version changes pulse peakiness values. The authors should therefore verify their lead and ice detection rates (see above).

The PP and SSD thresholds are taken from Laxon et al. (2013), though there is a scaling factor of 100 difference present in the PP. As the PP was not explicitly deifned in Laxon et al. (2013), the values are calculated following Armitage and Davidson (2014). (This citation was included on P7L27 but not on P7L30 nor P8L8 – this has been corrected in the manuscript, section 4.3). Following Kurtz et al. (2014), we have resampled the baseline-c data so that each data type (SAR and SARin) are consistent with 128 range bins per shot. See above for lead/ice detection rates.

P8L8: Same PP threshold magnitude inconsistency. Laxon et al. 2013 reports PP < 9 and SSD > 4 as criteria for sea ice surface returns.

See answer above.

P8L33ff: Does this mean that the authors have removed the DAC and tides from the CryoSat-2 derived elevations and then applied a consistent DAC, tide and mss correction for both ATM and CryoSat-2? What about the inverse barometric correction (see comment above)? This approach however still leaves the ionospheric and tropospheric corrections as an uncertainty factor, which may a dynamic range of 10 cm or more (Ricker et al Remote Sensing, 2016) thus a non-negligible magnitude. A second validation in form of along-track freeboard might help to improve the initial validation.

To make the absolute elevation comparison, we are following the method put forth by Yi et al. (2018), which "…replaced the ocean tide, ocean loading tide, and the inverse barometer correction used in the CryoSat-2 Baseline C data product with ocean tide, ocean loading tide, and dynamic atmospheric correction…". Yi et al. (2018) was not published at the time of submission, but has since been cited in this manuscript. This was done to have consistent geophysical corrections between the CryoSat-2 and ATM data. The ionospheric and tropospheric corrections from the CryoSat-2 data product were applied.

An along-track freeboard comparison would help to improve the initial validation, however, IceBridge freeboard from these campaigns has not yet been processed by the OIB team. We have added the correlation coefficients and further explanation to this paragraph, section 5.

P9L19ff: Filtering waveform is standard practice. The question is how many are filtered out?

This filtering method results in the filtering of just under 42% of the total number of waveforms. The average valid waveform fraction is .5832, which has been added to this paragraph in the manuscript, section 6.1.

P9L24f: The practice of computing freeboard by subtracting monthly means of sea surface heights and surface elevation is definitely not state-of-the-art. It puts a lot of trust in the range and tidal corrections that to my experience is not justified. I strongly

suggest to estimate freeboard orbit-wise, which also gives a better handle to identify
sea surface height estimation issues.

Thank you for the suggestion. We have modified our freeboard computation process slightly.
However, a true on-orbit calculation produced artifacts in the data (specifically in the Ross sea)
and therefore we elected not to use this method. Instead, we computed a monthly mean sea
surface height grid and used on-orbit surface elevations to find freeboard. We take each surface
elevation point (on-orbit) and subtract the corresponding sea surface height value to provide a
measure of freeboard for each data point. This provided a better option for filtering incorrect
retrievals. The remaining freeboard values are then gridded on a 25km grid. While a true on-
orbit calculation would indeed be better, we feel this method is sufficient for demonstrating the
retrieval results at this time.

P9L29: The description is a bit confusing, as the earlier sentences reads as snow
freeboard (per grid cell) = mean elevation (per grid cell) - mean ssh (per grid cell). From
this sentence I get the impression that the authors compute a mean sea surface height,
remove that from the all elevations, filter anomalous elevations and then compute the
snow freeboard from the remaining values. Please revise.

Agreed. This paragraph has been updated to reflect the new freeboard calculation technique.

P9L30: The authors seem to filter negative freeboards within a grid box. I assume this
are then from individual waveforms? The filter should be lower than 0 meter or else
the negative part of the range noise distribution will be filtered out for thinner ice and
thus cause the freeboard to be biased high. In the ESA CCI dataset, the filter range for
CryoSat-2 along-track freeboard is -0.25 to 2.25 meters.

Thanks for the comment. We have expanded the filter range to include negative freeboards, but
have chosen a range of -0.1m to 2.1m, which has been included in the text in section 6.1.

P9L34: Is the 125km filter applied to reduce noise, or to interpolate between gaps?

Mainly, the filter is applied to reduce noise, but it also interpolates between gaps caused by
filtered/missing data. The sentence "Smoothing is applied to reduce noise in the CryoSat-2 data
and also to fill gaps in the data" has been added to the manuscript, section 6.1.

P10L6: The impact of surfaces waves into the ice pack also needs to be considered for
CryoSat-2 data.

Thank you for pointing this out – the effects of surface waves has been included in the
manuscript, as well as some possibilities for seeing high freeboard here (like lower CS-2 data
density, and differing ice types found in the Antarctic MIZ). This can be found in section 6.1,
paragraph 2.

P10L17ff: For this comparison it would be helpful to have corresponding maps of surface
type (lead/ice) detection rates as well as average resnorm values to look into these
differences and verify that the sub steps of the CryoSat-2 snow freeboard algorithm are
working as intended.

This has been added to figure 5 in the manuscript (see above).

P10L16: Is the ICESat data also filtered to an effective resolution of 125 km?
That is correct – the ICESat data are filtered using the same method.

P10L26ff: It is good that the authors looked into the other output parameters of the waveform fitting. Unfortunately in this shorted form, more question are raised than answered. In essence, the retracking is based on a backscatter model of snow that when applied to CryoSat-2 waveforms, does not result in realistic snow conditions. Again, additional information such as regional differences in surface type detection rates or waveform fit quality parameters would greatly help to identify potential issues.
Verification of waveform fit quality and surface type detection has been included in the manuscript, figure 5. That said, it is not certain (at least from these figures) what is causing the unrealistic snow conditions. In the East Antarctic sector, where the "snow depth" was largest, there appears to be a larger fraction of floe-type waveforms. However, it is not anomalously large (and no larger than what is viewed in the Weddell Sea, and even smaller than the MIZ). There is an interesting region of invalid waveforms in the area, but it is small compared to the "snow depth" distribution found here, and not present elsewhere where the "snow depth" is large (such as further up the coast away from the Ross Sea). Likely, the snow-ice interface is not being tracked well, and instead found to be between 10-15 cm below the air-snow interface everywhere except the East Antarctic sector. More work will have to go into assessing the snow-ice interface elevation in future study.

P11L34: This is an overstatement given that all existing studies that used CryoSat-2 for Antarctic sea ice are not referenced in this paper.
This statement was included with intent to show the novelty of retrieving snow freeboard from CryoSat-2, though you are correct that more studies need to be referenced. We have included references to more studies throughout the paper, including in the revised introduction and here in the conclusion. This particular statement has been revised to better reflect the current state of the literature, referencing the CryoSat-2 work done by Schwegmann et al. (2016) and Paul et al. (2018), found in section 8.

P19Fig6: axis annotations and ticks are difficult to read on a printed version
This figure has been remade and should be easier to view in printed form – thank you.

[revised manuscript text omitted]

---

## Author Response (AR2)

We thank Reviewer #2 (Stefan Hendricks) and the Editor (Ted Maksym) for the additional comments to the manuscript. The responses, as well as the revised document showing the tracked changes, can be found below.

**Reviewer #2 comments (Stefan Hendricks):**

The authors have made a commendable effort to address my comments and significantly improved their paper compared to the original submission. The spectrum of scientific literature is now reflected properly and they provide more information to judge the skill of the snow freeboard retracker. The authors have refrained from computing snow freeboard on a per orbit basis but have given compelling reasons for this decision. I only would have liked to see a more detailed sensitivity study, not of impact of the initial guess, but on potential ambiguities between surface roughness and snow backscatter. But I see that this would inflate the paper quite a bit and I therefore recommend this to the authors for a follow on study.

In general I am satisfied and recommend this paper for publication, give that a few minor points are addressed (see specific comments).

Best Regards,
Stefan

Stefan,
Thank you so much for the constructive comments. You can find the responses and changes made below, in blue. We will continue to study the surface roughness/snow backscatter relationship in this method, and include our findings in follow-on studies of this work.

Specific Comments.

P3 L2 ff:
The main difference between radar freeboards and freeboard is indeed the absence of any range corrections. The most common correction is however the wave speed propagation correction for EM waves in snow that do not travel at vacuum light speed. The magnitude of the correction is approximately 20% of the thickness of the snow layer when the main backscatter horizon is located at the snow/ice interface. In this case, radar freeboards will always be lower (not higher) than freeboard since the range is based on a too large velocity. This effect will also counteract positive snow backscatter biases.

Paul et al. only implemented a correction for snow wave propagation and not snow backscatter, as the spatio-temporal variability of this effect is poorly understood.
Thanks for the comment and clarification. We have removed the part saying that radar freeboards likely track a layer that is higher than the actual snow ice interface. The sentence now reads:

"Freeboard retrievals that neglect range corrections for radar propagation through a snow layer are referred to as "radar freeboards".

We have also updated P3L4 to read:
"To counteract the effects of the snow layer on electromagnetic wave propagation, Paul et al. (2018) included…"

P3 L25:
Add the reason for the limitation (missing vertical resolution)
Added the range resolution (0.234 m, which is half of the range resolution in vacuo prior to taking the power envelope of the signal (0.468 m) (following Jansen, 1999)). P3L25 now reads: "…limited by the receive bandwidth (320 MHz, corresponding to a vertical resolution of 0.234m) and therefore not…. "

P10 L15:
To my knowledge there is no working sea ice type algorithm from OSI-SAF in the southern hemisphere. It is true that product files exist for the southern hemisphere, which indicate "young first year sea ice" for the day and region of the overflight. The authors are therefore correct in their statement, but I do wonder if this is an error in the ice type product (wrong flag). According to the product user manual sea ice type in Antarctica should always be "ambiguous":

_Note, that at present OSI SAF gives no information on ice type in the Antarctica. This is due to that the Antarctica sea ice classes has still not been studied enough to be included in the algorithm. Therefore, in the OSI SAF sea ice type product delivered for SH all the sea ice is classified as "ambiguous"._ [Excerpt from product user guide]

I would therefore suggest caution using the OSI-SAF data in the southern hemisphere. This is however also not a critical issue for this paper.
Thanks for the info. You're right – it appears that the files for these specific dates may have errors in the flags. We have removed this line altogether to avoid any confusion.

**Editor Comments:**

Comments to the Author:
Based on the favorable reviews of the revised manuscript from both referees, I am pleased to accept the manuscript for publication in The Cryosphere after addressing the few very minor comments of reviewer #2. In addition, based on my own reading of the revised manuscript and initial comments by the reviewers, I would suggest a couple other minor edits to the manuscript text to tone down some of the conclusions. The results in this paper are preliminary, based on necessarily limited comparisons that while promising, do compare rather poorly in some regions. As the authors themselves suggest further exploration of the sensitivity of the waveform fitting method is needed (and as stated will be undertaken). A few additional

qualifying statements in a few places might be helpful so that the limitations of the method (in present form) are clearer to the reader. For example:

We thank the editor for the detailed review of the manuscript, and appreciate these additional comments. Our responses and changes made to the manuscript can be found below, in blue.

The sentence on line 19-20 of the abstract is a strong statement, and does not convey any potential weakness in the method. e.g., figure 10 shows large differences with ICESat in some regions, the reasons for which are unclear, and while figure 7 shows a promising agreement overall, there are substantial differences over long segments that seem unlikely to be explained by the differences in footprint. A sentence that points out such weaknesses is warranted here.
Agreed. We have modified P1L19 to read:
"While our results suggest that this physical model and waveform fitting method can be used to retrieve snow freeboard from CryoSat-2, allowing for the potential to join laser and radar altimetry data records in the Antarctic, larger (~30 cm) regional differences from ICESat and along-track differences from ATM do exist, suggesting the need for future improvements to the method."

Figure 6 shows some reasonably large sensitivity to the fitting method. This is acknowledged. But perhaps the authors could add a sentence or two that conveys the magnitude of the potential impact of this uncertainty on the overall results.
Thanks for the suggestion. The end of the last paragraph of Section 4, which discusses figure 6, now reads:

"In this case, the effect of altering the backscatter parameterization had a larger effect on freeboard than altering $\sigma$. It is evident that physically inconsistent initial guesses can result in altered freeboard distributions, with the magnitude of the impact potentially being large (broad-scale difference of ~25 cm in Fig. 6 C). While this uncertainty surely adds to that of the overall results, the use of physically consistent first guesses acts to reduce the uncertainty as much as possible. Thus, a future area of study will be…"

Page 13, line 10-12 (and last sentence in conclusions). This is speculative, and I am skeptical that this is positive evidence that snow depth is being more accurately retrieved here. This region can have positive freeboards, but I don't think there is evidence to suggest it is more common here than elsewhere (such as the western Weddell), and there certainly has been flooding observed here. To be fair, the authors do not make a firm positive statement here, but I'd qualify it a bit more.
Fair point – we have added more to the last sentence of this paragraph, discussing the fact that realistic snow depths are not also seen in other areas of positive freeboard (e.g. the Western Weddell) which shows the need for more work.
It reads:
"Furthermore, the fact that these snow depth measurements are not higher over other areas of known positive ice freeboard, such as the western Weddell Sea, could signal issues regional issues of the algorithm to retrieve the snow-ice interface."

A similar point has been added to the conclusions (P14L7):
"More work is needed to understand why this region shows near-realistic snow depths while other regions with similar ice characteristics (e.g. positive ice freeboard in the western Weddell Sea) do not."

Figure 10 – as noted above, there are major differences with ICESat in some areas. Some of these differences may be due to the different time periods as noted, but it is interesting that Cryosat-2 freeboards are significantly lower than ICESat freeboards in all areas that typically have deep snow. Could this suggest a potential area for future improvement of the waveform model?
Definitely – we have added the following line to the conclusions/future work section (P13 L35) to address this point:
"The fact that the largest differences between CryoSat-2 and ICESat occur in regions of known thick snow depths could signal a difficulty of the algorithm over the thickest snow, suggesting an area for future improvements to the model."

Minor technical corrections:

Page 1, line 29 – I think you mean higher (i.e. further south) latitudes here, rather than lower.
You're correct - Lower was meant to mean the lowest (most negative) latitudes. We've updated this to read "south polar latitudes" to avoid confusion.

Page 1, line 32 – 'balance' doesn't seem like the right word (the energy budget will balance regardless of the albedo). Perhaps 'moderate'?
Changed to "moderate".

Page 2, line 38 – '..higher than the snow-ice...'
Corrected – thank you!

[revised manuscript text omitted]